# Drought as a possible contributor to the Visigothic Kingdom crisis and Islamic expansion in the Iberian Peninsula

Jon Camuera [1] ✉, Francisco J. Jiménez-Espejo [1] ✉, José Soto-Chica[2], Gonzalo Jiménez-Moreno [3], Antonio García-Alix [3], María J. Ramos-Román[4], Leena Ruha[5,6] & Manuel Castro-Priego[7]

The Muslim expansion in the Mediterranean basin was one the most relevant and rapid cultural changes in human history. This expansion reached the Iberian Peninsula with the replacement of the Visigothic Kingdom by the Muslim Umayyad Caliphate and the Muslim Emirate of Córdoba during the 8th century CE. In this study we made a compilation of western Mediterranean pollen records to gain insight about past climate conditions when this expansion took place. The pollen stack results, together with other paleohydrological records, archaeological data and historical sources, indicate that the statistically significant strongest droughts between the mid-5th and mid-10th centuries CE (450–950 CE) occurred at 545–570, 695–725, 755–770 and 900–935 CE, which could have contributed to the instability of the Visigothic and Muslim reigns in the Iberian Peninsula. Our study supports the great sensitivity of the agriculture-based economy and socio-political unrest of Early Medieval kingdoms to climatic variations.

Throughout human history, rises and falls of empires, kingdoms, and countries have been controlled by economic, social, and political factors, some of them partially conditioned by climate changes[1,2], including the recent conflicts in the Middle East[3,4]. The study of past environmental changes in a specific area is essential to understand how climate could have affected the political stability of a region. Nowadays, climate is becoming a central element in explaining historical crises in the Mediterranean area, as has occurred in recent decades around the Dark Age Cold Period and its relationship with the appearance of the Justinian´s plague[5], or with the crisis that affected rural settlements in some parts of Europe during those centuries[6]. One of the most interesting periods, and one that remains relatively unknown from a climatic point of view, is the period of Islamic expansion in North Africa[7], culminating in the conquest of the Iberian Peninsula. The formation of al-Andalus (Muslim-ruled area in the

Iberian Peninsula) from 711 CE led to the rapid disappearance of the Visigothic Kingdom, in a process that combined military conquest, political crisis, territorial fragmentation, and pacts[8,9]. The Visigothic Kingdom ruled the Iberian Peninsula for around 300 years (5th–8th centuries CE), replacing the Roman Empire rule, until the beginning of the Islamic expansion in Iberia during the early-8th century CE. This decline was marked by a civil war and the invasion of the Muslim Umayyad Caliphate (Islamic dynasty) in 711 CE until the final conquest in 726 CE. Although the archeological sites studied during the last decades of the Visigothic period provide useful information about farming strategies, urbanisms, territorial control or social organization[10], the rapid fall of the Visigothic Kingdom is still under debate, and the hypothesis of climate triggering the Islamic expansion from the northern Africa to the northern Iberia has not been investigated yet[11,12].

[1]Andalusian Earth Sciences Institute (IACT), Spanish National Research Council - University of Granada (CSIC-UGR), Granada, Spain. [2]Department of Medieval History and Historiographic Sciences and Techniques, University of Granada, Granada, Spain. [3]Department of Stratigraphy and Paleontology, Faculty of Science, University of Granada, Granada, Spain. [4]Faculty of Education, Mid-Atlantic University, Madrid, Spain. [5]Natural Resources Institute Finland, Oulu, Finland. [6]Research Unit of Mathematical Sciences, University of Oulu, Oulu, Finland. [7]Unit of Archaeology, Department of History and Philosophy, University of Alcalá, Alcalá de Henares, Spain. ✉e-mail: Jcamuera@gmail.com; Francisco.jimenez@csic.es

To date, the scarcity of both high-resolution paleoclimate records and studies focusing on the impact of climate changes in Early Medieval kingdoms during the 5th–10th centuries CE has prevented an assessment of whether climate changes controlled the decline of the Visigothic Kingdom and the Islamic expansion in Iberia. Previous Holocene paleoclimatic studies at decadal- to centennial scale showed that the main climate factor affecting the environmental variability in the Mediterranean region is the amount of precipitation, which has strongly affected the human population and dispersal in this area[13,14]. Here we used a high-resolution pollen stack based on 107 pollen records from Iberia and Morocco (Fig. 1) along with other paleohydrological records, archeological data and historical sources to show how the persistent droughts, identified under the Scale-normalized Significant Zero crossing (SnSiZer) statistical analysis, could have influenced the economic, social and political conditions during the Visigothic Kingdom and al-Andalus between the 5th and 10th centuries CE.

## Results and discussion

### Reconstruction of drought periods from Iberian/Moroccan paleoenvironmental proxies

The stacks of the arid-adapted *Artemisia* and xerophyte taxa display an overall increasing trend over the last 5000 years, indicating a progressive aridification over time (Fig. 2e, f). In addition, the results of *Artemisia* at subdecadal-scale resolution allow the identification of 23 drought events during the last 5000 years (black arrows in Fig. 2e). Increases in *Artemisia* and xerophyte data, indicative of arid conditions in the Iberian Peninsula, occurred at ca. 4900, 4550, 4150, 3900, 3650, 3450, 3200, 2900, 2650, 2400, 2300, 2100, 1950, 1750, 1500, 1400, 1250, 1150, 1000, 850, 700, 500 and 200 cal yr BP.

Comparison of pollen data with isotopic analysis of speleothems provides a more detailed insight into local environmental and regional paleoclimate changes in the study region[15,16]. In addition, since human influence on speleothem records is lower than on the vegetation records, the comparison between these two proxies allows us to corroborate whether the aridity periods observed in pollen records are predominantly natural or not. Changes in speleothem carbon and oxygen isotopes can be driven by different factors[16], including vegetation density[17], cave temperatures[18], or the amount of precipitation, among others[19]. The correlation analyses between the drought events recorded by the pollen data and the high-resolution oxygen and carbon isotopic records from caves in northern Morocco (Chaara cave, composite $\delta^{18}O$)[20] and northern Spain (Cobre, Kaite, and Mayor caves, stacked relative $\delta^{13}C$)[21] show high positive correlation ($r = 0.80$ between *Artemisia* and $\delta^{18}O$, $p = 0.007$; $r = 0.76$ between *Artemisia* and $\delta^{13}C$, $p = 0.010$; for further details, see Methods) (Fig. 2c–e). This positive correlation suggest that the Moroccan and Iberian caves could have been controlled by similar paleohydrological conditions. The increasing $\delta^{18}O$ and $\delta^{13}C$ speleothem values from Morocco and Spain suggest a reduction in precipitation, which although subjected to small age uncertainties, coincides with the drought periods recorded by the stacked *Artemisia* record. Similar $\delta^{18}O$-palaeohydrological patterns have been observed in speleothems and in recent meteoric water records from Iberia, showing increasing $\delta^{18}O$ values during decreasing precipitation conditions[22,23]. The high $\delta^{13}C$ values during drought periods could reflect a lower contribution of the light carbon isotope in the speleothem as a consequence of decreasing vegetation density surrounding the cave[19,24]. These periods of sparser vegetation would agree with increasing *Artemisia*, and therefore, more steppe taxa (Fig. 2c, e, f). However, other effects resulting in high $\delta^{13}C$ during arid conditions cannot be ruled, such as the increase of $CO_2$ degassing of the dripwater during low rainfall periods (low infiltration) and low cave drip rates[25], or a reduction of the soil biogenic $CO_2$ production via microbial activity and plant respiration, which were also related to low humidity conditions[16].

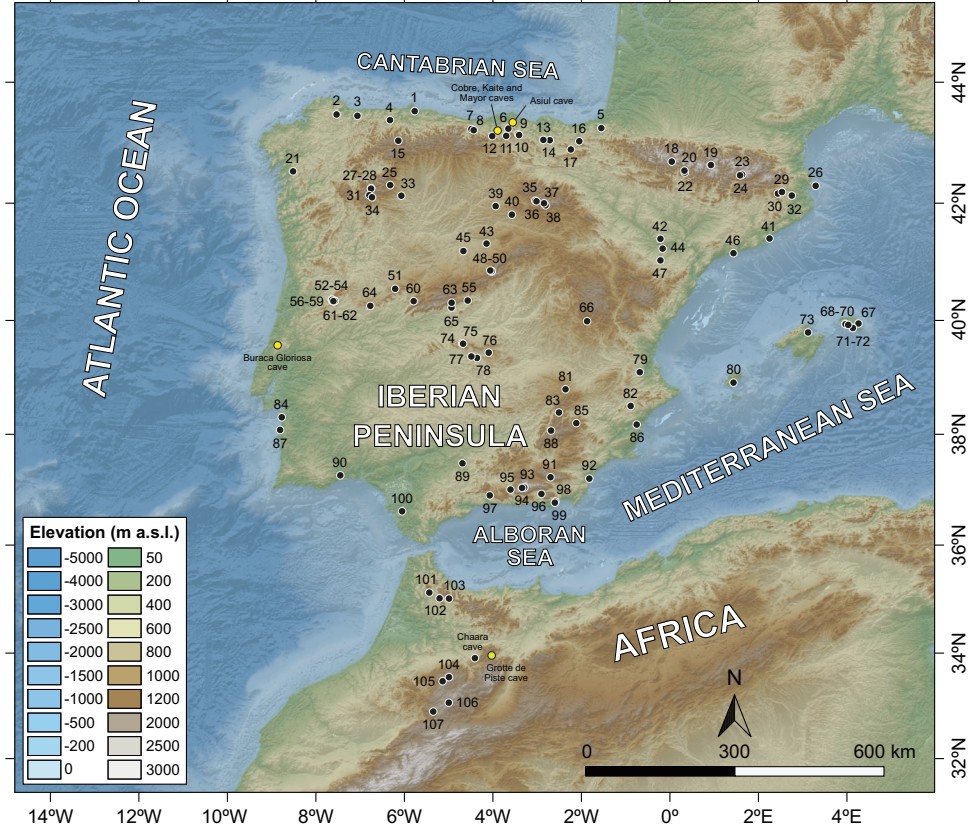

**Fig. 1 | Location of the 107 fossil pollen records (black dots) compiled and used in this study.** Yellow dots show the Iberian and Moroccan cave records cited in the manuscript and used for comparison with the pollen results. For detailed information of the fossil pollen records, see Supplementary Data 1.

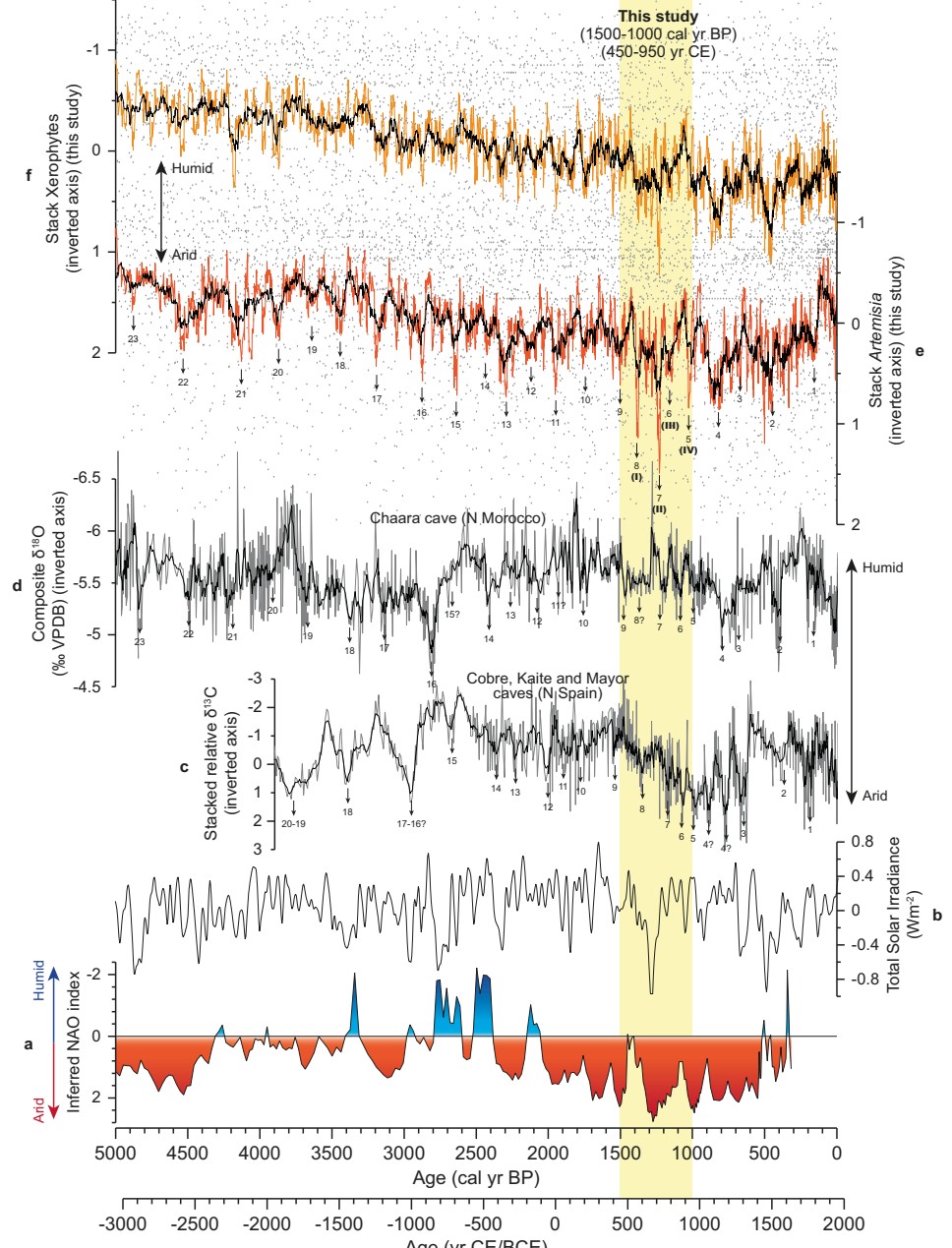

**Fig. 2 | Comparison of climate proxies/mechanisms for the last 5000 years.**
**a** NAO index reconstruction (inverted axis)[83]. **b** Total Solar Irradiance (W/m²)[84]. **c** Stacked relative $\delta^{13}C$ record (inverted axis) from Cobre, Kaite, and Mayor caves (N Spain)[21]. **d** Composite $\delta^{18}O$ speleothem record (‰ VPDB, inverted axis) from Chaara cave (N Morocco)[20]. In graphs **c** and **d**, the gray lines show the raw relative $\delta^{13}C$ and $\delta^{18}O$ values, and the black lines the 5-point moving averages. **e–f** Stacks of *Artemisia* and xerophytes (inverted axes) from this study, with black lines representing the 45-point moving averages, and the orange (*Artemisia*) and yellow (xerophytes) lines the 15-point moving averages. The extreme droughts recorded by the *Artemisia* data, and observed in the Spanish $\delta^{13}C$ and Moroccan $\delta^{18}O$ records, are indicated with black arrows. In graph **e**, the droughts I, II, III, and IV occurring during the period of interest for this study (450–950 CE, 1500–1000 cal yr BP, yellow vertical shade) correspond to the drought events 8, 7, 6 and 5 of the last 5000 years. The source data from graphs **e** and **f** are available in Supplementary Data 2.

The period between 450 and 950 CE (1500–1000 cal yr BP) deserves special attention, as it presents four arid events, one of them presenting the maximum aridity peak of the last 5000 years at ~700 CE (1250 cal yr BP), shown by the maximum abundance of *Artemisia* and xerophytes (Fig. 2e, f). The performed SnSiZer statistical analysis of the stacked *Artemisia* record between mid-5th and mid-10th centuries CE (450–950 CE, 1500–1000 cal yr BP) identified four statistically significant extreme drought periods at: (I) 545–570, (II) 695–725 CE, (III) 755–770 CE, and (IV) 900–935 CE (Fig. 3a, b), corresponding to the drought events 8, 7, 6, and 5 of the last 5000 years (Fig. 2e). The previously mentioned $\delta^{18}O$ (N Morocco) and $\delta^{13}C$ (N Spain) records

also showed four events of climatic deterioration between 450 and 950 CE (Fig. 2c, d). Global-scale dendroclimatological studies showed that the coldest climate conditions of the last two millennia occurred at 536–565 CE[26] in connection with three main volcanic eruptions at 536, 540, and 547 CE[27], matching with our statistically significant drought period I (Fig. 3a, b). In addition, the speleothem records from Asiul (N Spain)[28] and Buraca Gloriosa (Portugal)[29] caves presented significant increases in $\delta^{18}O$ values at ca. 540–580 and 700–750 CE and at ca. 500–600 and 700–770 CE, respectively, consistent with the most arid conditions identified in the pollen data during the drought periods I and II (Fig. 3a, b and Supplementary Fig. 1c–e). Another speleothem

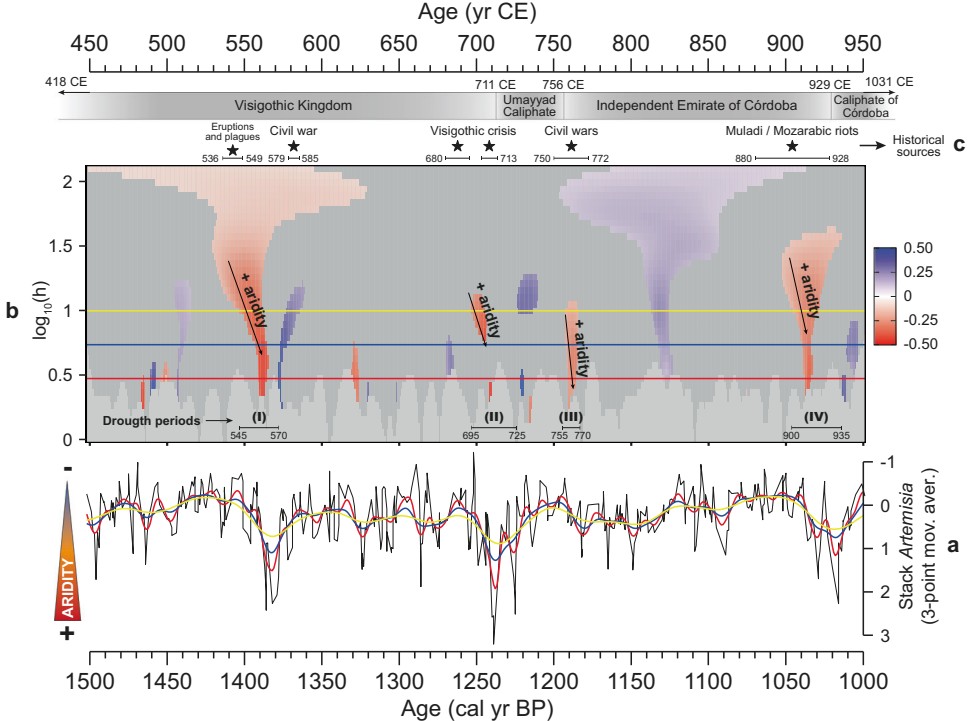

**Fig. 3 | The SnSiZer analysis of the *Artemisia* stack between 450 and 950 CE (1500–1000 cal yr BP) to identify the strongest drought periods along with the events collected from historical sources. a**, **b** The stack of *Artemisia* (3-point moving average, inverted axis) and the SnSiZer graph with three smoothing levels (red, blue, and yellow lines in both panels). In the SnSiZer graph (**b**), red and blue colors suggest strong increasing and decreasing trends of *Artemisia*, respectively. The numbers I, II, III, and IV (bottom of **b**) indicate the four statistically significant drought periods identified by the SnSiZer (age ranges included). **c** Events and duration of the compiled historical sources (black stars and horizontal lines). The source data from graph **a** are available in Supplementary Data 2.

record from Morocco, the GP5 speleothem record from Grotte de Piste cave[30], which recorded the environmental conditions between 600 and 2010 CE, presented two hiatus surfaces linked to arid conditions at ca. 620–750 and 900–940 CE. These two events of stalagmite growth cessation match relatively well with three statistically significant drought periods shown by the statistical analysis of the stacked *Artemisia*, the first hiatus coinciding the droughts at 695–725 and 755–770 CE (drought periods II and III), and the second hiatus with the one at 900–935 CE (drought period IV).

A recent quantitative precipitation reconstruction from the Iberian Peninsula[31] recorded an annual precipitation decline of 0–100 mm/yr at 700–800 CE for northern Iberia and of 200 mm/yr at 680 CE for southern Iberia (Supplementary Fig. 1f). Moreover, the palynological study from Medina lake[32] and the lake level reconstruction of Zoñar lake[33], both sites located in southern Iberia, showed similar severe drought conditions during the late-7th and early-8th centuries CE. The analysis of non-pollen palynomorphs from Medina lake of the last 5000 years recorded the highest abundance of *Gelasinospora* spore at ca. 710 CE, which has been linked to dry lake conditions[34,35], and is coeval with the presence of *Artemisia* throughout the entire record, reinforcing the interpretation of low rainfall conditions during the early-8th century (Supplementary Fig. 1g). In Zoñar lake the lowest lake level of the last 2500 years occurred at ca. 650–750 CE, which could also indicate that the lowest precipitation conditions of the last two and a half millennia were recorded during the late-7th and early-8th centuries CE (Supplementary Fig. 1h).

### Relationship between the four statistically significant drought periods and historical sources between mid-5th and mid-10th centuries CE (450–950 CE, 1500–1000 cal yr BP)
In the following sections, we show the relationship between the four SnSiZer statistically significant droughts observed in the *Artemisia*

pollen record at (I) 545–570, (II) 695–725, (III) 755–770 CE and (IV) 900–935 CE, and the different historical events collected from historical sources between mid-5th and mid-10th centuries CE (Fig. 3a, b). In recent years, the contributions of written historical sources for the Visigothic period (5th–8th centuries CE) have been supported by archeological data and, in particular, by studies on inhabited rural areas and subsistence resources[36]. Some archeological results have also allowed us to distinguish different forms of peasant habitats (villages, farms) and their permanence or disappearance over time[37], whereas other studies have provided new data about the agricultural production strategies[38], including research on the expansion of sheep and goat farming, and changes in crop management[39,40]. All this took place in a changing political context, especially from 633 CE onwards, and with a reconfiguration of production systems and land ownership, which would have been closely linked to climate and environmental changes.

### Drought period I (545–570 CE)/Eruptions, plagues, and Visigothic civil war
The first drought period recorded by the Iberian and Moroccan *Artemisia* and xerophyte data between 450 and 950 CE, and statistically significant under the SnSiZer, occurred at 545–570 CE (Figs. 2e, f and 3a, b).

The historical sources, such as the Byzantine historian Procopius, noted that during 536–538 CE in the Italian peninsula, inclement weather caused severe crop damages and unusually harsh and long winters that lasted through the springs and summers, as well as cases of cannibalism due to famine[41]. The Roman statesman Cassiodorus wrote that *"The heat from the sun was feeble"* and *"A winter without storms, a spring without mildness, and a summer without heat"*[42], leading to prolonged frosts and droughts. This climatic situation, which seems to have been triggered by the above-mentioned volcanic eruptions[27], could have promoted food shortages and famine, the

onset of the Late Antique Little Ice Age (LALIA), and the outbreak of the Justinian plague in 541–543 CE through the Mediterranean[2,26,43]. The chronicle of Victor of Tunnuna stated that this plague affected Hispania in 542 CE[44]. Later on, the Gallo-Roman historian Gregory of Tours wrote that during 580/581 CE the ambassadors of the king of Neustria returned from Hispania with news about plagues and famine that were ravaging Carpetania (regions surrounding the Tagus valley) and spreading to Septimania (SE of present-day France)[45]. The historian also wrote that during 580–585 CE Hispania suffered strong frosts and droughts, coinciding with the Visigothic civil war (580–584 CE) (Fig. 3c). Considering these historical sources and the pollen results, the statistically significant drought period I recorded at 545–570 CE (Fig. 3a, b), which roughly coincides with the onset of the LALIA and probably linked to the volcanic eruptions and the cold climate conditions[2,27], could be considered an important factor for the spread of the famine and plagues during the mid-6th century CE in Iberia. After the drought period I (545–570 CE), the Visigothic state under king Leovigild (Visigothic king, reign 568/569–586 CE) was able to consolidate a policy of centralization, territorial expansion, and strengthening of the fiscal system after a long period of monetary devaluation[46]. It is likely that the climate amelioration could have contributed to an improvement of agriculture and, therefore, of the kingdom administration and stability, which could also have contributed to a better economic condition for the construction of new urban centers, such as Reccopolis (578 CE), Victoriaco (581 CE) and Oligicus (621 CE).

## Drought period II (695–725 CE)/Visigothic Kingdom crisis and Umayyad Caliphate expansion

The *Artemisia* and xerophyte stacks from Iberia and Morocco show that the maximum drought conditions of the last 5000 years occurred at 695–725 CE, indicated by the highest *Artemisia* values and identified by the SnSiZer analysis (Figs. 2e and 3a, b).

The vegetation records from the Spanish Central System showed decreasing tree cover, especially in montane areas, and increasing fire activity, which was related with the effect of human activity to obtain new pastures and lands for farming[47,48]. However, the changing climate conditions forced the cultivation of cold-adapted rye (*S. cereale*) at high altitudes, suggesting that although some vegetation changes in pollen records were affected by human activity[47], these variations could have been forced at the same time by cold and drought conditions. Actually, during the Visigothic period, different techniques were developed in rural areas to adapt to drought conditions. In southern Iberia (Sierra Nevada range) they built the oldest aquifer recharge systems in Europe, dated at $636 \pm 80$ CE[49], and population moved to higher elevations with the consequent increase in population in mountain regions of southern Iberia[50]. This population displacement to highlands took advantage of the higher altitudinal rainfall conditions compared to lowlands[51]. In addition, the archeological evidence from the Visigothic city of Reccopolis (central Iberia), such as the construction of terraces and the reorganization of the agricultural system, have also been related to forest decline and dry conditions during this period[52].

Historical sources suggest the beginning of persistent drought conditions at around 680 CE in the Iberian Peninsula. This is coeval with the increasing aridity after ca. 670 CE in Anatolia and the Levant, which caused the abandonment of agricultural land and the collapse of the major settlements[53]. The XII Council of Toledo, organized in central Iberia by the king Ervigio (Visigothic king, reign 680–687 CE), took place in November 681 CE *"to support a world in collapse"*[54]. Three years later, in November 684 CE, Title III of the Minute Book of the XIV Council of Toledo recorded: *"This note, with all that has happened, was presented in the worst moment of a terrible weather, when all lands, in the worst winter moment, are covered by heavy snowfall and glacial temperatures"*[55], pointing into harsh

winter conditions during the Autumn season. In addition, the Mozarabic chronicle of 754 CE stated that during this period: *"Ervigio was consecrated in the kingdom of the Goths. He ruled for seven years, and Hispania suffered a huge famine"*[56]. During the next reign of king Egica (Visigothic king, reign 687–702/703 CE), in particular during the years 691–692 CE, a bubonic plague that affected the Iberian Peninsula[56] was related to long periods of cold/drought conditions, low food productivity and famine[57–59] (Fig. 3c). The Mozarabic chronicle also recorded that during 702–703 and 710–713 CE *"The starvation and plague ravaged Hispania"*[56]. Moreover, the chronicles point to periods of severe famine (707–709 CE) immediately prior to the period of conquest (711 CE)[60] and the Arabic Akhbār majmūʿa chronicle (mid-11th century) indicated that around half of the population of Hispania died of starvation between 706 and 710 CE[61,62] (Fig. 3c). These chronicles point to a climate change towards more arid and colder conditions during late-7th and early-8th centuries CE. The death of king Wittiza (Visigothic king, reign 700–710/711 CE) in 710/711 CE, the consequent civil war, and the Muslim invasion of the Visigothic Kingdom (711 CE) took place during the identified persistent *Artemisia* pollen drought period II (695–725 CE) (Fig. 3b, c). In this regard, both the chronicle of king Alfonso III (Christian king, reign 866–910 CE) and the Mozarabic chronicle of 754 CE coincided about the decline of the kingdom during the invasion: *"The Goths perished part by hungry and part by the sword"* (chronicle of Alfonso III)[63] and *"With sword, famine and captivity, not only Hispania Ulterior but also Hispania Citerior was devastated"* (Mozarabic chronicle)[56]. Note that the increase in *Artemisia* at 695–725 CE shows a delay with respect to the first historical sources on harsh climate conditions at 680–684 CE. This 15-year delay could be related with age model limitations and/or the resilience of vegetation biomes prior to the expansion of arid-adapted *Artemisia* in the region.

Despite the described harsh climate conditions, the archeological records do not show massive devastation or destruction of cities or settlements during the invasion[64]. In the Early Medieval kingdoms at least 60% of the richness was based on agricultural products. Agricultural labor employed around 80% of the population[65], and therefore, harsh and arid climate conditions would have affected the stability of kingdoms. However, in the case of the Visigothic Kingdom, there is no unanimous position on the main factors that could have led to the collapse of the Visigothic state[66]. The decline of the Himyarite kingdom and the emergence and expansion of Islam during the 6th century CE across the Arabian Peninsula has recently been related with arid climate conditions and strong droughts[67]. This drought recorded in Arabia as well as the one we recorded in Iberia during late-7th and early-8th centuries CE must also have affected the North African regions. These arid conditions constitute a new element in political, economic, and military motivations that led to the Islamic expansion of the Iberian Peninsula from 711 CE onwards. For several decades, there has been a consensus that Islamic expansion from the Middle East and North Africa to the Iberian Peninsula may have been favored by the introduction and diffusion of new technologies, agricultural practices, and products better adapted to drylands and arid environmental conditions[68]. However, there is increasing evidence for the diffusion of pre-Islamic agricultural techniques, tools, and crops adapted to semi-arid regions, suggesting that the pre- and post-Islamic Near East and Mediterranean areas were more similar to each other than previously assumed[69].

Although the Visigothic social and political crisis could have started prior to the drought period II recorded by the pollen data, we can conclude that the extreme and persistent drought during the late-7th and early-8th centuries CE in the Iberian Peninsula may have had a negative impact on food production, damaging the primarily agriculture-based economy and pastoral activities[70], triggering (at least partially) the social and political instability that could have

affected the decline of the Visigothic Kingdom, and the first phases of the consolidation of al-Andalus.

## Drought periods III and IV (755–770 and 900–935 CE)/Islamic riots and civil wars

The drought period revealed by the SnSiZer analysis during the increase in *Artemisia* at 755–770 CE due to enhanced aridity in the region (drought period III, Fig. 3a, b), coincides with the first Muslim political destabilization after the establishment of the Umayyad Caliphate in 711 CE.

Muslims attempted to expand northward, but were stopped by the Franks at the Battle of Poitiers (France, 732 CE)[71]. Later on, in 740 CE, the Berber uprising that started in the North Africa and in al-Andalus/Hispania generated several civil wars, famine, and depopulation of certain areas, promoting the rupture of al-Andalus with the Abbasid Caliphate (Islamic dynasty) and becoming an independent Emirate (Emirate of Córdoba) in 756 CE[72]. According to the Mozarabic chronicle, during the years 748–750 CE *"A horrible famine desolate the entire al-Andalus/Hispania"*, similar to that recorded by the Akhbār majmūʿa chronicle for the 750–754 CE *"The year of the Hegira 132 / 750 CE, God sent a major drought and famine to the entire al-Andalus/Hispania"* and the chronicle of Moissac (11th century) for the 771-772 CE *"A devasting famine affect all Hispania"*[73] (Fig. 3c). Both the stacked *Artemisia* record and the chronicles show drought conditions during the mid-8th century CE. The beginning of Abd al-Rahman's rule (Muslim king, reign 756–788 CE) was characterized by several sociopolitical crises, which could have been a consequence of the drought and subsequent famines, and only from 783-784 CE onwards shows an increase in monetary and expanding territorial control, linked to the stabilization of taxation[74].

The last strong drought period identified by the SnSiZer analysis of pollen data occurred at 900–935 CE (drought period IV, Fig. 3a, b). This last episode coincides in part with the first *fitna* (Muslim civil war), a long process of political crisis that began with the most powerful rebellion led by Omar Ibn Hafsun from 880 to 928 CE[75] (Fig. 3c). These confrontations have been linked to the expansion of the process of conversion to Islam and the increase of the political and territorial control of the Emirate of Córdoba[76].

The two statistically significant increases in the stacked *Artemisia* record and historical sources suggest extreme drought conditions during the mid-8th century and the early-10th century CE. These droughts could have affected agriculture, contributing to low food production and famines, and thus promoting the economic and sociopolitical destabilization of al-Andalus. However, this process of macro analysis cannot ignore the fact that communities responded to climatic and environmental conditions. In this regard, the Andalusian society was characterized by the expansion of new agricultural techniques, including irrigation[77].

## Droughts in the western Mediterranean region and possible climate mechanisms

Numerous paleoclimate studies have demonstrated a strong relationship between internal atmospheric dynamics, such as the North Atlantic Oscillation (NAO), and external solar activity, including Total Solar Irradiance (TSI)[78,79]. The NAO index controls the strength and direction of the westerlies over Europe and thus the amount and source of precipitation entering the western Mediterranean[80]. Although the NAO reconstruction for the Iberian Peninsula[81] does not match well with the pollen data between 450 and 950 CE (Supplementary Fig. 1a, c), the most arid conditions shown by the increase in *Artemisia* and xerophytes at the end of the 7th and the beginning of the 8th centuries CE occurred during the highest positive North Atlantic NAO index reconstruction[82,83] and the lowest TSI[84] of the last 5000 years (Fig. 2a, b, e, f). The low solar irradiance generates cold and fresh North Atlantic Current conditions, affecting the state of the NAO[85].

However, this relationship has been a topic of debate over the last decade. Some studies have shown that a reduced solar irradiance during the Holocene may have modulated the negative NAO phases[20,86], whereas other proxy records have evidenced a relationship between low solar irradiance and arid conditions in the Mediterranean[87–89], which could be related to positive decadal- to centennial-scale NAO phases. In this latter case, a low solar irradiance could have produced a southward movement of the polar front and a reduction in the advection of water masses in the Atlantic, reducing the storm activity over the Mediterranean and resulting in arid climate and drought conditions[88]. An exception occurred during the cold and drouth period of the mid-6th century CE, interpreted as a possible consequence of three volcanic eruptions occurring in a short time period (536, 540, and 547 CE)[27] and coeval with the Justinian plague.

## Methods
### Fossil pollen data

This study synthesized pollen and archeological data from Spain, Portugal, Andorra, and Morocco. With respect to the pollen data, we focused on the analysis of *Artemisia* and other xerophyte pollen taxa from lacustrine continuous sedimentary records to gain insight into the aridity crises of the last 5000 years and, in particular, of the last 1500–1000 cal yr BP (450–950 CE). The fossil pollen data acquisition was performed using the Neotoma database (www.neotomadb.org) and the European Pollen Database (EPD, www.europeanpollendatabase.net), whereas other pollen records not included in these databases were obtained by privately contacting the authors of the corresponding studies (see *References* in Supplementary Data 1). We selected all the continental pollen records from Spain, Portugal, Andorra, and Morocco containing data-samples during the last 5000 years, which resulted in 107 pollen records (Fig. 1). In order to make the study as objective as possible, none of these records have been filtered according to the number and/or precision of dates. We consider that filtering the data based on the number of dates and/or their precision would cause subjectivity in the data acquisition and, therefore, results would lose credibility.

The stacked *Artemisia* record is based on 3977 samples whereas the xerophyte record is based on 4115 samples. Note that the number of samples in the *Artemisia* and xerophyte records is different due to the absence of *Artemisia* in seven records. The detailed information (*Site no. map, Sitename, Database, Site ID, Dataset ID, Latitude, Longitude, Altitude, Chronology source, Dating and depths, Age model details,* and *References*) of the 107 pollen records are included in Supplementary Data 1, and the stacked *Artemisia* and xerophyte data in Supplementary Data 2.

### Age chronologies

Age chronologies for the pollen records were obtained from databases, including the chronologies from the original studies and the chronologies produced by the Mapping and Data Accuracy working group (MADCAP) of the EPD[90,91]. As the calibrated age difference between the different calibration curves for the period of interest (450–950 CE, 1500–1000 cal yr BP) is negligible (<20 calibrated years, see Supplementary Fig. 2), we decided to use the original calibration curves (i.e., IntCal98, IntCal04, IntCal09, IntCal13 and IntCal20) provided by the databases and the authors (Supplementary Data 1). Since each of the fossil records collected for the stacked *Artemisia* pollen curve has its own age model, it is not possible to provide specific age uncertainties for this composite record. This problem has been mitigated by collecting a large number of fossil records (107 records), which reduces the age uncertainties.

### Pollen as a proxy for precipitation

Although the increasing abundance of *Artemisia* and other xerophyte taxa (such as Amaranthaceae and *Ephedra*) are well known as

indicators of aridity periods in the western Mediterranean during the Quaternary[92–94], we made a calibration using the Eurasian Modern Pollen Database (EMPDv2)[95] and the WorldClimv2.1 data (www. worldclim.org)[96] to observe the relationship between high percentages of *Artemisia* and low precipitation conditions. Therefore, the recent mean annual precipitation (mm/yr) and *Artemisia* percentages from 8217 modern pollen sites were used to analyze the relationship between the abundance of *Artemisia* and the precipitation values (Supplementary Fig. 3). The modern pollen database shows that sites with precipitation values below ca. 400 mm/yr present mean *Artemisia* percentages over 7%, whereas the mean *Artemisia* percentages decrease below 3% with precipitation values above ca. 400 mm/yr, suggesting that the *Artemisia* pollen genus is a good proxy for interpreting arid climate conditions in this region. Therefore, persistent droughts in past periods caused an expansion of the fast-growing and arid-adapted herb/grass species, such as *Artemisia*, resulting in a higher proportion of these taxa in the fossil records.

The limitations of the pollen analysis in paleoclimate interpretations have to be taken into account, since modifications in the vegetation cover caused by human impact could also have affected pollen abundances. However, it is more likely that the human effect on vegetation could have had an impact during a given period in a particular area of the Iberian Peninsula and/or Morocco, but it is less likely that this effect would have affected different large Iberian/Moroccan areas at the same time. Therefore, in order to reduce the human effect on vegetation, we have collected a large number of pollen records over a large territory, reducing the noise of human disturbance that could have affected some pollen records during specific periods.

## Data normalization
A z-score normalization (x − mean/stdev) was performed on the fossil *Artemisia* and xerophyte data (individually for each record) before stacking the data, resulting in high-resolution *Artemisia* and xerophyte stacks for the last 5000 years (Fig. 2e, f). A 45-point and 15-point moving averages were performed on the *Artemisia* and xerophyte stacks (black lines and orange/yellow lines in Fig. 2e, f) for an easier comparison with other paleoclimatic records. The normalized stacked *Artemisia* and xerophyte records are included in Supplementary Data 2.

## Correlation analyses
A Pearson correlation analysis was performed between our stacked *Artemisia*, the composite $\delta^{18}O$ from Chaara cave (Cha1 and Cha2 speleothems, N Morocco), and the stacked relative $\delta^{13}C$ from Cobre, Kaite and Mayor caves (C11, LV5, and SLX1 speleothems, N Spain). The stacked *Artemisia* (last 5000 cal yr BP), composite $\delta^{18}O$ (last 5000 cal yr BP), and stacked relative $\delta^{13}C$ (last 3900 cal yr BP) records present a data resolution data of 1.3, 4, and 4.5 years, respectively. Before performing the correlation analyses, *Artemisia* and composite $\delta^{18}O$ data were interpolated (linear interpolation) to the lowest data resolution of the $\delta^{18}O$ record, i.e., 4 years, whereas *Artemisia* and stacked relative $\delta^{13}C$ were interpolated to the lowest data resolution of the $\delta^{13}C$ record, i.e., 5 years. The results show a high positive correlation, with $r = 0.802$ ($p = 0.007104$) for the stacked *Artemisia* vs composite $\delta^{18}O$ records, and $r = 0.768$ ($p = 0.01058$) for the stacked *Artemisia* vs stacked relative $\delta^{13}C$ records. The correlation analysis was developed under the Past 4.12b statistical software[97].

## Scale-normalized Significant Zero crossing (SnSiZer) analysis
The Significant Zero crossing (SiZer) analysis is a useful tool in paleoecology to assess the relevant thresholds in the data and to define where statistically significant changes occur in a data set[98]. In this analysis, nonparametric smoothing is applied to data using a sequence of smoothing levels in order to reveal salient features in the signal at all frequencies. The Scale-normalized Significant Zero crossing (SnSiZer) analysis[99], based on the original SiZer analysis[100], has

been used to observe trends and anomalies in the data, allowing the identification of statistically significant increases and decreases in the data. In addition, the SnSiZer uses scale-normalized derivatives[101] to detect the statistically significant changes in the signal and highlight the intensity of increasing and decreasing changes (red and blue), resulting in more or less intense colors in the graph. Thus, in the SnSiZer graph, reading from past to present (i.e., from left to right), red and blue areas indicate increases and decreases of *Artemisia*, respectively (note the inverted axis in Fig. 3a), and therefore, relevant increasing and decreasing aridity trends. Dark gray areas represent no significant changes, whereas the light gray areas indicate that the sampling resolution is too low (Fig. 3b). When a certain level of smoothing (red, blue, or yellow horizontal lines in the vertical axis, Fig. 3b) cuts the relevant increase (red) or decrease (blue) in *Artemisia*, it means that those changes are strong under that smoothing level. The recently developed SnSiZer was implemented by modifying the source code of the sizer R-package[102]. The data was preprocessed using a 3-point moving average smoother to reduce background noise in the stacked *Artemisia* record and the error of the age chronologies. However, such pre-smoothing may produce autocorrelation in the data and cause false positives in the SnSiZer analysis. In order to mitigate the autocorrelation effect, the SnSiZer analysis was run under the 95% confidence level.

To define the age ranges of drought periods, we took the onsets of the *Artemisia* increases (red areas in the SnSiZer graph, Fig. 3b), which indicate the beginning of the droughts, until the onsets of the decreases (blue areas in the SnSiZer graph), indicating the end of drought conditions. The maximum resolution for determining the onsets/ends of droughts is 5 years. Since the end of the third strong drought period (drought period III) is not characterized by an abrupt decrease in *Artemisia* (and therefore, not observed in the SnSiZer graph at the selected smoothing levels −red, blue, and yellow lines−), this drought was delimited by the strong increase in *Artemisia* identified by the SnSiZer analysis between 755 and 770 CE (Fig. 3b). Therefore, this methodology between 450 and 950 CE resulted in four statistically significant droughts at 545–570 (1405–1380 cal yr BP), 695–725 (1255–1225 cal yr BP), 755–770 (1195–1180 cal yr BP) and 900–935 CE (1050–1015 cal yr BP). Although the statistical significance of these events cannot be warranted due to the above-mentioned risk of false positives, they can be considered as stronger droughts in the data.

## Historical and archeological sources
Historical sources under annual/bi-annual resolution were collected and compared with the pollen and paleoclimate proxy data to understand the climate impact on the Visigothic and Muslim societies. The historical sources were selected according to the geographical and chronological criteria and the reliability of the source, including the Roman stateman Cassiodorus (6th century), the Gallo-Roman historian Gregory of Tours (6th century), the Roman-Byzantine historian Victor of Tunnuna (6th century), the Minute Books of the XII and XIV Councils of Toledo (7th century), the Mozarabic chronicle of 754 CE (8th century), the Christian chronicle of king Alfonso III (9th century), the Arabic Akhbār majmū'a chronicle (11th century) and the chronicle of Moissac (11th century). In addition, a review of the most significant archeological data has been carried out. Special emphasis has been placed on rural sites of the Visigothic period. Data from urban sites from the period between the 6th and 8th centuries CE have also been analyzed.

## Reporting summary
Further information on research design is available in the Nature Portfolio Reporting Summary linked to this article.

# Data availability
The data from this study, including the detailed information on all the Iberian and Moroccan fossil pollen records and the *Artemisia* and

xerophyte stacks are available in Supplementary Data 1 and 2, respectively. The *Artemisia* and xerophyte stacks (Supplementary Data 2) are also available at the Pangaea data repository (www. pangaea.de). The raw fossil pollen data were obtained from the open-access Neotoma database (www.neotomadb.org) and the European Pollen Database (www.europeanpollendatabase.net), and by contacting the authors privately (Viñuelas, Brezoso, Medina, Zoñar, Padul, Río Seco and Hondera records, see *References* in the Supplementary Data 1). The seven pollen records are not available in open databases, can be obtained by contacting the following authors: C. Morales-Molina (Viñuelas and Brezoso), T. Schröder (Medina), C. Martín-Puertas and P. González-Sampériz (Zoñar), J. Camuera (Padul), R.S. Anderson and G. Jiménez-Moreno (Río Seco), and J.M. Mesa-Fernández (Hondera). The source data from this study that were used for Figs. 1–3 and Supplementary Fig. 1 are available in Supplementary Data 1 and 2.

## Code availability

The SnSiZer code is a modified version of the original SiZer code published by Chaudhuri and Marron (1999)[100], and developed by modifying the source code of the sizer R-package (Sonderegger, 2020)[102]. Since the new SnSiZer code has not yet been published, the code will be available privately upon request.

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

## Acknowledgements

We thank all the contributors of pollen records from the Neotoma database and the European Pollen Database. We also thank Cesar Morales-Molina, Tabea Schröder, Celia Martín-Puertas, and Penélope González-Sampériz for providing the pollen data from Viñuelas, Brezoso, Medina, and Zoñar. This work is funded by the Ministry of Science and Innovation of the Government of Spain, *Agencia Estatal de Investigación /10.13039/501100011033/ and Fondo Europeo de Desarrollo Regional - A way of making Europe,* in particular the grant numbers FJC2020-044215-I (author J.C.), *PID2021-125619OB-C21* (authors G.J.-M. and A.G.-A.) and *PID2021-125619OB-C22* (author F.J.J.-E.). The grant number *Retos P20_00059* (author G.J.-M.), the action *Proyectos I + D + i del Programa Operativo FEDER 2018* (grant number *A-RNM-336-UGR20*, authors G.J.-M., and A.G.-A.) and the research group *RNM-190* from the *Junta de Andalucía* (Regional Government of Andalusia). The project *SBPLY/21/180501*/000205 (author M.C.P.) from the Scientific Research and Technology Transfer Projects of the *Junta de Comunidades de Castilla-La Mancha* (Regional Government of Castilla-La Mancha).

## Author contributions

J.C. and F.J.J.-E. designed the study. J.C. collected the pollen data and performed the data processing and statistical analysis. L.R. developed the code of the SnSiZer analysis based on the original SiZer and provided the code to perform the analysis. The SnSiZer code will be freely provided upon request. J.S.-C. collected information about historical sources. J.C., F.J.J.-E., and M.C.-P. analyzed the archeological data. J.C., F.J.J.-E., J.S.-C., G.J.-M., A.G.-A., M.J.R.-R., L.R., and M.C.-P. wrote the manuscript.

## Competing interests

The authors declare no competing interests.
