## [Peer Review File · Nature Communications]

Droughts contributed to the Visigothic Kingdom crisis and Islamic expansion in the Iberian Peninsula (5th to 10th centuries CE)Reviewers' Comments:

Reviewer #1:

Remarks to the Author:

This is a very complex paper to review for a number of reasons. First of all, the main hypothesis (that the crisis of the Visigothic kingdom, the Islamic expansion in the 8th century and the subsequent crisis of the Andalusian state are a direct consequence of droughts) is complex and polemic enough to require more than a six page discussion. This is a very discussed topic in current historiography that would imply a detailed discussion in itself. In this regard, the paper does not take into account the recent and immense archaeological and documentary literature which may refute the main assertion of the paper. Just one little example, isotopic analysis made throughout the Iberian peninsula does not show any general famine or structural economic problem. Thus there is a lack of a crossed discussion with other sources that may reinforce (or refute) such a polemic statement.

Perhaps this is one of the biggest concerns with the paper, i.e. it takes the Visigothic kingdom as a solid state unity that "rises" and "falls" because of climatic changes. Evidence suggests that precisely in the end of the 7th century political and economic regionalization was very deep and some may consider that the crisis of the Visigothic kingdom was prior to the moment of this peak of droughts in the Iberian peninsula. Then it is possible that droughts may be aggravating a prior situation more than causing (something that is not being considered in depth in the paper). This also works for the 8th century expansion of the berber/arab army, why did the droughts affect the Visigothic state apparatus but not the Andalusian one? Why then the Andalusian state decided to stop the conquest? Why the same droughts were a problem for the Andalusian state in the 10th century AD but not in the 8th or 9th centuries. Again, this is a problem of the polemic character of the paper, which tries to give a single cause to very complex historical and political processes which would require much more discussion to become a solid argumentation or, at least, a solid hypothesis to be explored in the future.

There is also a serious problem with the instrumental use of the documentary sources. There is kind of an arbitrary selection of them. Famines, droughts and economic problems were very common between the 5th and the 10th century and they are not only documented for this period (just two example, Gregory of Tours writes about these sort of problems for the Iberian peninsula in the 6th century AD; and the Justinian plague also affected the Iberian peninsula during the first decades of the 6th century). Why specifically droughts and famines in the period considered in the paper did effectively create a political crisis? Again, a complex discussion of the political and economic situation that surrounded those droughts (which in fact would require more extension).

A last big concern relates with the methodology used. I have to admit that I am not a pollen specialists, being archaeology my background. But I would need a more detailed discussion on the dating of these samples to really understand the relationship between droughts through artemisia and xerophyte data. Authors themselves recognise that "only recently, the Visigoths have been focus for scientific research, but the archaeological evidences are still poorly dated compared with other Iberian and Roman periods", using to support this assertion a paper on the Visigothic slates (which does not discuss the chronological problem for the whole Visigothic period) and a paper summarizing the historiography of the topic. However, the contexts of the 107 pollen records, the specific dating of those records and a deeper discussion on how they specifically relate to the periods considered are not present and, from my point of view, are crucial for the main discussion of the topic.

I have more concerns and problems with the paper but all of them can be summarized in the idea that more discussion and evidence coming from other sources (mainly archaeological sources) is needed to really support a controversial assertion such as the one defended in the paper.

However, I want to finish the review considering that the data and the effort to gather all the pollen analyses is really interesting and may raise quite compelling ideas and debates, but the way it is proposed is just polemic, more suitable for a general journal than in a scientific paper. Thus I will suggest a reformulation of the paper into a more subtle suggestion and a deeper discussion on the specific ways in which droughts may have affected the economy of the Iberian peninsula in a regional basis, considering other sources. This is something present in the paper in page 5 when authors writes that "We can therefore conclude that the extreme and persistent drought event during the late 7th

and early 8th centuries in the Iberian peninsula COULD HAVE affected food production and primarily agriculture-based economy, affecting (AT LEAST PARTIALLY) the social and political stability that MAY have caused the decline of the Visigothic Kingdom"; it is precisely in the discussion of this possibility and hypothesis, and not taken as a definitive conclusion, where the interest of the paper may lie.

Reviewer #2:

Remarks to the Author:

Establishing firm and sound relationships between climatic, environmental and socio-cultural changes is a challenging task and there is always the risk of climate determinism. The manuscript presents a stacked pollen record from over hundred sites in Spain (and Morocco) to investigate the impact of late Holocene droughts on the Visigothic Kingdom, the Umayyad caliphate and the Muslim Emirate of Cordoba. The stacked pollen record has a high-temporal resolution, allowing to resolve decadal-scale droughts. However, it is not clear how this record was constructed and how accurate the stacked record is on such short-timescales as almost all pollen records have a low-resolution, not to mention their chronological uncertainties. The fundamental question is: How can you use low-resolution records to develop a single one with a much higher resolution? The information on the statistical approach are somewhat vague and it is therefore impossible for almost all readers to evaluate the accuracy of the stacked pollen record on such short timescales. Furthermore, what are the chronological uncertainties of the stacked record? The authors should add supplemental information to show the development of the stacked pollen record in greater detail. In its current state, this appears to be a "black box" for most readers.

The comparison between the pollen record and historical is quite convincing, but the quality of the manuscript could be greatly improved by a longer and more detailed discussion on the specific socioeconomic impacts of droughts on the Visigoths, the caliphate and Emirate of Cordoba. I would suggest to delete the paragraph of the potential causes of the droughts to have more space available to move beyond the rather simplistic relationship between drought and collapse/change.

Overall, the manuscript is very interesting and potentially suited for Nature Communications after major revisions were made. I would be willing to review the revised version of this manuscript.

Reviewer #3:

Remarks to the Author:

REVIEW OF THE MANUSCRIPT ID NCOMMS-22-45602-T: 'Droughts contributed to the Visigothic Kingdom crisis and Islamic expansion in the Iberian Peninsula (7th to 10th centuries)' by Camuera et al.

This paper constitutes a very interesting contribution to Paleoclimatology with regional to global implications and provides new evidence on the potential influence of climate crisis on historical changes. In particular, the authors aim to demonstrate the influence of intense drought episodes on the Islamic expansion in the Iberian Peninsula and internal socioeconomic crises in Iberian Muslim kingdoms during the subsequent centuries. The climatic context of these historical changes has been relatively understudied compared to other relevant cultural changes, making this research timely and novel. Furthermore, this study integrates existing proxy data from pollen records of the Iberian Peninsula and Morocco into a high resolution hydroclimatic reconstruction able to register short-lived droughts during the last millennia, following an innovative methodological approach.

Therefore, this work deserves publication in the journal 'Nature Communications.' However, there are several issues that need to be addressed. Most of the points raised in the comments are of moderate importance and do not affect the main statements of the paper.

Main points

1) The use of Artemisia pollen percentages as direct evidence of arid conditions remains controversial

for recent chronologies (late Holocene) as this proxy has been also related with anthropogenic disturbances. In the Methods section (lines 252-256), the authors indicate that, according to the modern pollen database, *Artemisia* pollen percentages occur in percentages over 7% in regions below 400 mm/yr. However, the sensitivity of this proxy in the record of short-lived (decadal and subdecadal timescales) drought episodes needs further discussion. A potential correlation with other independent (i.e., non-pollen based) hydroclimatic reconstructions of comparable temporal resolution, such as recent instrumental and available tree ring records in the Iberian Peninsula and Morocco (Esper et al., 2007) and nearby regions for overlapping chronologies might contribute to reinforce the validity of this proxy.

2) The structure of the discussion on the main three droughts reconstructed in this research is unbalanced in the manuscript. A well-documented, detailed comparison with other regional and global palaeohydrological records is made for the first crisis (695-725 CE). In contrast, the last two episodes are exclusively discussed in terms of socio-economic consequences and historical background. The paleoclimatic context of the last two episodes should be also discussed with a similar degree of detail. As an alternative, the two paragraphs between lines 80 and 109 might be moved to the end of the paper, after the historical description of the three aridity crises, incorporating a comparison of obtained results with other paleoclimatic records of the region for the three aridity crises described in this paper.

3) The *Artemisia* and xerophyte stacks presented in this paper are based on the whole dataset of available Iberian and NW African records. However, in the discussion (lines 78-94), these records are only correlated here with Southern Iberian paleoclimate records (Medina and Zoñar lake sediment sequences). A comparison with the precipitation reconstruction for Northern Iberia (available from Ilvonen et al. (2022)) and other terrestrial records is also required here to understand the consistency between pollen dataset and these reconstructions, at the same spatial scale. If drought episodes reconstructed in this research have been recorded (or not) with variable intensities in Northern Iberia, it should be also discussed here. In addition, in lines 95-109, the comparison of reconstructed droughts with the rainfall reconstruction for the Western Mediterranean region based on speleothem records from Morocco carried out by Ait Brahim et al. (2019) is relevant here and it should be also included in Extended Data Fig. 1 and discussed here. A potential comparison with the Buraca Gloriosa speleothem record (Western Portugal) (Thatcher et al., 2020), covering a shorter period (i.e., the last ca. 1200 years) might be also useful for the last drought discussed in this manuscript. In contrast, the correlation with Eastern Mediterranean records should be taken with care as significant differences in the hydrological evolution of Western and Eastern areas of the Mediterranean basin during the last millennium have been reconstructed at these timescales in relation with NAO forcing (Roberts et al., 2012). Although the aridity crisis reconstructed in this research are previous to 900 CE, these potential hydrological contrasts should be taken into account in the discussion (lines 108-109).

4) The description of potential climatic mechanisms driving reconstructed aridity crises in the Iberian Peninsula are not sufficiently clear and lack detail. The relationship between arid and humid phases at centennial scale in the Western Mediterranean Region (e.g., Medieval Climate Anomaly and Little Ice Age) with NAO variability has been already discussed and evidenced in a number of paleoclimate works. However, what is the specific mechanism for these short-lived aridity crises occurring at decadal timescales? A more specific discussion on these potential mechanisms is required here.

Minor points

Lines (L) 25-27) This sentence is unclear. I understand that, according to the authors, persistent droughts occurred during this period might have contributed to the destabilization of Muslim kingdoms ruling the Iberian Peninsula. Therefore, 'internal rebellions and civil wars' were a consequence of climate conditions and I suggest replacing 'due to' with 'promoting'.

L 34-36) The chronology of Islamic expansion in the circum-Mediterranean area is needed here.

L 36-38) The end of Visigothic Kingdom is repeated here and this sentence is a bit unclear. Please,

replace 'until its fall during the early 8th century when' with 'the beginning of Islamic expansion in Iberia.'

L39) I think '(Islamic dynasty)' might be deleted here.

L42) Please, replace 'have' by 'has'

L42-43) Although the number of archaeological studies of Visigothic settlements is comparatively lower than others in the region, this sentence is too conclusive. Please, rephrase.

L45-46) There is a considerable number of terrestrial, high-resolution records in Spain covering the Middle Ages, compared to other periods (e.g., late glacial). In fact, the authors cite most of these works and have used pollen data for this research. Please, rephrase.

L74) 'Is' instead of 'are'.

L194) A new, specific North Atlantic Reconstruction (NAO) for the Iberian Peninsula is available (Hernández et al., 2020) and it is probably more adequate for its comparison with reconstructed hydroclimatic conditions and the discussion of potential driving mechanisms for the investigated aridity crisis. I also suggest incorporating this reconstruction to Fig. 2 and Extended Data Fig. 1.

REFERENCES

- Ait Brahim, Y., Wassenburg, J. A., Sha, L., Cruz, F. W., Deininger, M., Sifeddine, A., . . . Cheng, H. (2019). North Atlantic Ice-Rafting, Ocean and Atmospheric Circulation During the Holocene: Insights From Western Mediterranean Speleothems. *Geophysical Research Letters*, 46(13), 7614-7623. doi:<https://doi.org/10.1029/2019GL082405>
- Esper, J., Frank, D., Büntgen, U., Verstege, A., Luterbacher, J., & Xoplaki, E. (2007). Long-term drought severity variations in Morocco. *Geophys. Res. Lett.*, 34(17), L17702. Retrieved from <http://dx.doi.org/10.1029/2007GL030844>
- Hernández, A., Sánchez-López, G., Pla-Rabes, S., Comas-Bru, L., Parnell, A., Cahill, N., . . . Giralte, S. (2020). A 2,000-year Bayesian NAO reconstruction from the Iberian Peninsula. *Scientific Reports*, 10(1), 14961. doi:10.1038/s41598-020-71372-5
- Ilvonen, L., López-Sáez, J. A., Holmström, L., Alba-Sánchez, F., Pérez-Díaz, S., Carrión, J. S., . . . Seppä, H. (2022). Spatial and temporal patterns of Holocene precipitation change in the Iberian Peninsula. *Boreas*, 51(4), 776-792. doi:<https://doi.org/10.1111/bor.12586>
- Roberts, N., Moreno, A., Valero-Garcés, B. L., Corella, J. P., Jones, M., Allcock, S., . . . Türkeş, M. (2012). Palaeolimnological evidence for an east-west climate see-saw in the Mediterranean since AD 900. *Global and Planetary Change*, 84-85, 23-24. doi:10.1016/j.gloplacha.2011.11.002
- Thatcher, D. L., Wanamaker, A. D., Denniston, R. F., Asmerom, Y., Polyak, V. J., Fullick, D., . . . Hays, J. A. (2020). Hydroclimate variability from western Iberia (Portugal) during the Holocene: Insights from a composite stalagmite isotope record. *The Holocene*, 30(7), 966-981. doi:10.1177/0959683620908648

Reviewer #4:

None

The reviewers' comments are in **black** and our responses in **blue**.

When we refer to the lines (e.g., "Lines 177-182"), we are referring to the clean file without the *Track Changes* activated.

Reviewer #1 (Remarks to the Author):

This is a very complex paper to review for a number of reasons. First of all, the main hypothesis (that the crisis of the Visigothic kingdom, the Islamic expansion in the 8th century and the subsequent crisis of the Andalusian state are a direct consequence of droughts) is complex and polemic enough to require more than a six page discussion.

Thank you for the comment. However, several previous studies, published in short scientific articles such as this one, show that climate disturbances had a profound impact on human societies (e.g., Fleitmann et al., 2022). In our study we are describing the climatic scenario when the Islamic invasion and Andalusian riots took place, and we found that it coincides with extreme droughts in Iberia. We suggest that a harsh climate with strong droughts could have played a role in the demise of the Visigothic Kingdom, but we never stated that this was the only cause (see the title of the paper "Droughts contributed..."). We have modified the manuscript in order to make this statement clearer (Lines 177-182):

"Although the Visigothic social and political crisis could have started prior to the drought event recorded by vegetation data, we can conclude that the extreme and persistent drought during the late 7th and early 8th centuries in the Iberian Peninsula may have had a negative impact on food production and damaged the primarily agriculture-based economy, triggering (at least partially) and enhancing the social and political instability that could have affected the decline of the Visigothic Kingdom."

This is a very discussed topic in current historiography that would imply a detailed discussion in itself. In this regard, the paper does not take into account the recent and immense archaeological and documentary literature which may refute the main assertion of the paper. Just one little example, isotopic analysis made throughout the Iberian Peninsula does not show any general famine or structural economic problem.

Thanks for the suggestion. However, we would have been grateful if the reviewer had included references or studies that refute our main assertion ("Extreme drought was the climatic scenario when the Islamic invasion took place during early 8th century"), but he did not do so. We would be more than glad to include or discuss any reference or study that the reviewer knows based on archaeological and literature that refute our assertion.

With respect to the example about the isotopic analysis in archaeological remains, this is not correct. The isotopic analyses on archaeological samples (i.e., bones) could indicate changing diets (using $\delta^{13}\text{C}$ or $\delta^{15}\text{N}$) or migration (using $\delta^{18}\text{O}$), but they are not suitable to describe whether the individuals died or suffered from famine. In any case, we would be grateful if the reviewer could provide studies demonstrating that, based on isotopic data, an individual did (or did not) suffer from famine during a specific period.

Thus there is a lack of a crossed discussion with other sources that may reinforce (or refute) such a polemic statement.

We have included archeological and historical data, palynological, stable isotope data, and other paleoclimate records, demonstrating that this paper can be considered highly multidisciplinary. In any case, following the reviewer's comments, this new version of the manuscript includes more data and references than the previous version, which support and confirm our results.

Perhaps this is one of the biggest concerns with the paper, i.e. it takes the Visigothic kingdom as a solid state unity that "rises" and "falls" because of climatic changes. Evidence suggests that precisely in the end of the 7th century political and economic regionalization was very deep and some may consider that the crisis of the Visigothic kingdom was prior to the moment of this peak of droughts in the Iberian peninsula. Then it is possible that droughts may be aggravating a prior situation more than causing (something that is not being considered in depth in the paper).

We thank the reviewer for the interesting comment. We never said in the manuscript that the Visigothic Kingdom "rises" or "falls" were simply caused by climate changes. In the paper, we describe the climatic conditions occurring during the fall of the Visigothic Kingdom and the rise of the Umayyad Caliphate in Iberia. Based on the pollen data, the internal (economic, social and political) instability of the Visigothic Kingdom occurred during the first statistically significant extreme drought period, suggesting that the fall of the Visigothic kingdom could have been affected by different factors, being one of them the persistent drought climate conditions at that moment. However, as we previously said (and it has been made clearer in the new version of the manuscript), the extreme aridity at that time was not necessarily the only cause that produced the fall of the Visigothic kingdom. In fact, we have included that although the Visigothic social and political crisis could have started prior to the statistically significant drought recorded by the pollen data, this drought event could have had negative impact in the economy. This situation could have triggered/enhanced the social and political instability, contributing (at least partially) to the decline of the kingdom (Lines 177-182).

This also works for the 8th century expansion of the berber/arab army, why did the droughts affect the Visigothic state apparatus but not the Andalusian one?

This is also a very interesting comment. Droughts could have affected the Visigothic Kingdom more severely because the Visigothic economy was mainly based on agriculture (cereals), whereas in the less populated Berber regions of North Africa the economy was mainly based on pastoralism (goats, cattle). However, it is very likely that droughts also affected the northern African region because both areas are subjected to similar climatic conditions (i.e., Mediterranean climate), which could also have influenced the migration of the North African population towards northern latitudes, including the expansion towards the Iberian Peninsula. We have better clarified this in the new version of the manuscript (Lines 161-165). Unfortunately, the archeological-historical and paleoclimatic information available from North Africa is small, which limits the interpretation for the North African region.

If the reviewer is also referring to the influence of droughts on the Muslim population/army during the northward expansion and settlement of the Iberian Peninsula in the 8th century, it is very likely that it also affected them. However, the drought may have had a greater impact and influence on the socially and politically weakened Visigothic state than on the Muslim population/army in expansion. In addition, during this expansion, pastoralism may have contributed to the drought having less effect on the Muslim population that was expanding throughout the Iberian Peninsula.

Why then the Andalusian state decided to stop the conquest?

This subject is very interesting. However, the aim of this study is to describe the climate conditions during the Visigothic Kingdom crisis and the Islamic expansion in the Iberian Peninsula, not to discuss the reasons of the northern expansion of the Islamic conquests in Europe. Muslims were stopped by Franks in Poitiers (732 CE), but why they were stopped or why they did not continue the expansion to the north is not the goal of this paper.

Why the same droughts were a problem for the Andalusian state in the 10th century AD but not in the 8th or 9th centuries.

Thanks for the comment. This study shows that the drought intensity was higher in the 10th century than previously in the 8th-9th centuries. Actually, in our data and in the statistical analysis there is a lack of statistically significant drought events during the 9th century (Figure 3). The comparison of the statistically significant droughts occurring during the 8th century (at 755-770 CE) and the 10th century (at 900-935 CE) shows stronger arid conditions during the 10th century. Similarly, the stacked *Artemisia* values presents higher values at 695-725 CE (and therefore, more intense drought period) than during the other two previously mentioned drought events (755-770 and 900-935 CE).

Again, this is a problem of the polemic character of the paper, which tries to give a single cause to very complex historical and political processes which would require much more discussion to become a solid argumentation or, at least, a solid hypothesis to be explored in the future.

We understand the reviewer. However, in this paper we never pretended to be polemic giving a single cause to this very complex historical and political process. We just want to show that the pollen-based climate interpretations and the statistical analysis show that extreme arid events occurred during some crucial historical processes in Iberia (in this case, the Visigothic Kingdom crisis and the Islamic expansion, and the internal riots during the Umayyad Caliphate and the Emirate of Cordoba between the 7th and 10th centuries), indicating that climate could have played an important role (but not the only factor) during these extremely important historical events in Iberia.

There is also a serious problem with the instrumental use of the documentary sources. There is kind of an arbitrary selection of them. Famines, droughts and economic problems were very common between the 5th and the 10th century and they are not only documented for this period (just two example, Gregory of Tours writes about these sorts of problems for the Iberian peninsula in the 6th century AD; and the Justinian plague also affected the Iberian peninsula during the first decades of the 6th century). Why specifically droughts and famines in the period considered in the paper did effectively create a political crisis? Again, a complex discussion of the political and economic situation that surrounded those droughts (which in fact would require more extension).

This is a very interesting topic. We agree with the reviewer about what he/she says that Gregory of Tours wrote about the problems (famine, droughts) that occurred during the 5th and 6th centuries. In fact, our record also shows drought conditions during the period described by Gregory of Tours for the 5th and 6th centuries. The figure below is an expansion of the recent figure submitted to Nature Communications, but including the period between 450 and 650 CE (which was never included in the submitted version):

In this figure (as we previously said, never submitted to Nature Communications because we exceeded the length required for submissions to this journal), we can see that the SnSiZer analysis identified a strong drought occurred during the mid-6th century, which is the drought period that the reviewer talks about. In relation with this figure, in a previous version of the manuscript that we finally never submitted to Nature Communications, we wrote (among other historical sources we have):

“The historical sources show that in the year 592 CE, the Gallo-Roman historian Gregory of Tours wrote that during 579 CE the ambassadors of the king of Neustria returned from Hispania with news about plagues of locusts and famine that were ravaging Carpetania (regions surrounding the Tagus valley) and spreading to Septimania (SE of present-day France) (Gregorio de Tours, 2013). The historian also wrote that during 579-585 CE Hispania suffered strong frosts and droughts, coinciding with the Visigothic civil war (580-584 CE)”

Although in the recent version of the manuscript we have not included the period between 450 and 650 CE, we have indicated all the extreme droughts recorded during the last 5000 years:

*“Increases in Artemisia and xerophyte data indicative of arid conditions in the Iberian Peninsula occurred at ca. 4900, 4550, 4150, 3900, 3650, 3450, 3200, 2900, 2650, 2400, 2300, 2100, 1950, 1750, 1500, **1400**, 1250, 1150, 1000, 850, 700, 500 and 200 cal yr BP.”*

The period marked in **bold (ca. 1400 cal yr BP, 550 CE)** detected by our compiled pollen record (and identified by the SnSiZer analysis in the never submitted figure above), correspond to the severe drought that occurred during the 6th century.

Due to the length format of the Nature Communications, and the fact that the strongest droughts recorded in the last 5000 years took place just during the Muslim invasion of Iberia in the 8th century, we decided to focus only on the period between 650 and 950 CE. However, if the reviewers or the editor would find it convenient and interesting to include the period

between 450-650 CE, including both stacked pollen records and historical sources, we would be happy delve deeper into this period.

A last big concern relates with the methodology used. I have to admit that I am not a pollen specialist, being archaeology my background. But I would need a more detailed discussion on the dating of these samples to really understand the relationship between droughts through artemisia and xerophyte data. Authors themselves recognise that "only recently, the Visigoths have been focus for scientific research, but the archaeological evidences are still poorly dated compared with other Iberian and Roman periods", using to support this assertion a paper on the Visigothic slates (which does not discuss the chronological problem for the whole Visigothic period) and a paper summarizing the historiography of the topic.

Thanks for the comment. Pollen data do not originate from Visigothic archeological sites, but from continuous sedimentary records from lakes, bogs or caves. Paleoclimatic data from Visigothic and Muslim archeological sites are important but punctual, discontinuous and represent local conditions, which are not suitable for regional climatic reconstructions. That is why we selected fossil pollen data from continuous lacustrine records. All the available pollen records selected from the Iberian Peninsula (107 pollen records) are based on robust radiocarbon-dated age models, and all the details have been included in the Supplementary Tables 1 and 2, including details on the dating, age models and other useful information of each fossil pollen record. We have better clarified this in the Methodology section. We have also taken as a reference and included the pollen and archaeological study from the Visigothic city of Recópolis (central Iberia) (Olmo-Enciso et al., 2019), which coincide with our reconstruction.

However, the contexts of the 107 pollen records, the specific dating of those records and a deeper discussion on how they specifically relate to the periods considered are not present and, from my point of view, are crucial for the main discussion of the topic.

As we mentioned in the previous point, Rev#1 should keep in mind that the discussed pollen and paleoenvironmental records do not come from archaeological contexts, but from lacustrine continuous sections in the Iberian Peninsula, and cave speleothems (in the case of d13 and d18O records). This makes this chronology independent from the still poorly dated Visigoth remains, as Rev#1 acknowledged. We correlated the stacked pollen record based on the 107 Iberian and Moroccan pollen records with cave, lacustrine and other paleoclimate records, and the historical evolution of the Visigothic Kingdom and the Caliphate/Emirate. In the Supplementary Table S1 we show all the details ("sitename", "chronology source", "age model details", "references"...) related with each of the 107 pollen record from Iberia and Morocco. The age models of these records were independently established in the different studies based on sequential dates (mainly: 14C in the case of sediments, and U/Th in the case of cave speleothems). In addition to that, in the new Supplementary Table 2 we also included the normalized stacked *Artemisia* and xerophyte data. Since the raw data of 100 pollen records are in open access databases (i.e., Neotoma and European Pollen Database) and other authors have provided privately the raw pollen data from 4 records (we have acknowledged them in the section "Acknowledgements"), we have not uploaded the raw pollen data in a supplementary table. If the reviewers and/or the Editor want to verify the raw data, we have no problem to provide it privately. However, we think that including the four pollen records (plus the three records from Padul, Rio Seco and Hondera, which are records that we have previously studied) that are not available in open access databases in a supplementary table, would go against the privacy of the authors' data.

I have more concerns and problems with the paper but all of them can be summarized in the idea that more discussion and evidence coming from other sources (mainly archaeological sources) is needed to really support a controversial assertion such as the one defended in the paper.

Thanks for the comment. However, the reviewer fails to provide us with those “evidences coming from archeological sources”. We will be happy to discuss or reference any study with paleoclimatic information obtained in any archeological site in the Iberian Peninsula during the Visigothic period. During this revision, we have added information from further archeological sites related with the late Visigothic Kingdom. A recent paper published by Olmo-Enciso et al. (2019) indicate that the late 7th century and the last period of the Visigothic Kingdom was characterized by “forest decline and dry conditions during this period”. Therefore, according to some archeological sources, it seems that the end of the Visigothic Kingdom Iberian Peninsula was also characterized by arid conditions, as we expose in the manuscript with our pollen data and other paleoclimatic record from Iberia and Morocco.

However, I want to finish the review considering that the data and the effort to gather all the pollen analyses is really interesting and may raise quite compelling ideas and debates, but the way it is proposed is just polemic, more suitable for a general journal than in a scientific paper.

Thank you for your comments, but we still think that this work is suitable for a high-impact short-format scientific paper. We also believe that it can be a highly referenced paper and will provide with new perspectives on this period of huge historical impact in Europe.

Thus I will suggest a reformulation of the paper into a more subtle suggestion and a deeper discussion on the specific ways in which droughts may have affected the economy of the Iberian peninsula in a regional basis, considering other sources. This is something present in the paper in page 5 when authors writes that "We can therefore conclude that the extreme and persistent drought event during the late 7th and early 8th centuries in the Iberian peninsula COULD HAVE affected food production and primarily agriculture-based economy, affecting (AT LEAST PARTIALLY) the social and political stability that MAY have caused the decline of the Visigothic Kingdom"; it is precisely in the discussion of this possibility and hypothesis, and not taken as a definitive conclusion, where the interest of the paper may lie.

Thanks for the suggestion. In this new version of the manuscript we make clear that climate was probably not the only factor affecting the decline of the kingdoms/reigns (e.g., Lines 177-182), but climatic variations and droughts could have played an important role in social and cultural evolution of the Iberian Peninsula during the 7th and 10th centuries.

Reviewer #2 (Remarks to the Author):

Establishing firm and sound relationships between climatic, environmental and socio-cultural changes is a challenging task and there is always the risk of climate determinism. The manuscript presents a stacked pollen record from over hundred sites in Spain (and Morocco) to investigate the impact of late Holocene droughts on the Visigothic Kingdom, the Umayyad caliphate and the Muslim Emirate of Cordoba. The stacked pollen record has a high-temporal resolution, allowing to resolve decadal-scale droughts. However, it is not clear how this record was constructed and how accurate the stacked record is on such short-timescales as almost all pollen records have a low-resolution, not to mention their chronological uncertainties. The fundamental question is: How can you use low-resolution records to develop a single one with a much higher resolution?

The information on the statistical approach are somewhat vague and it is therefore impossible for almost all readers to evaluate the accuracy of the stacked pollen record on such short timescales. Furthermore, what are the chronological uncertainties of the stacked record? The authors should add supplemental information to show the development of the stacked pollen record in greater detail. In its current state, this appears to be a “black box” for most readers.

Thank you for the comment. We understand the concern about the chronological control and the age-model uncertainties. We have better clarified this in the new version of the manuscript in the “Methods” section, where we explain that the data corresponding to the different records were stacked to create two higher-resolution composite records based on the 100 *Artemisia* pollen records and 107 xerophyte pollen records from Iberia and Morocco. This is a common methodology when using the same paleoclimate indicator (e.g., composite/stacked record using speleothems: Martín-Chivelet et al., 2011; Thatcher et al., 2020; composite/stacked record using pollen records: Jiménez-Moreno et al., 2020; Pini et al., 2022) in regions with similar paleoclimate controls. In our case, since the *Artemisia* genus is a good indicator of aridity in the western Mediterranean (Carrión et al., 2010), a stacked *Artemisia* record using different *Artemisia* fossil pollen records from Iberia/Morocco is fully reliable to provide information about drought conditions in the Iberian Peninsula and northwestern Africa.

We understand the reviewer's concerns regarding the chronological uncertainties for each individual record. However, the large number of records used would reduce/compensate the age uncertainty of the stacked pollen record, providing a very high-resolution record with low age uncertainty. As also mentioned in “Methods”, the age uncertainty related with the different calibration curves (i.e., IntCal98, IntCal04, IntCal09, IntCal13 and IntCal20) for our period of interest (1300-1000 cal yr BP, 650-950 CE) is very low (<20 calibrated years), which does not influence the age chronology of the stacked *Artemisia* and xerophyte records.

We have also included the new “Supplementary Table 2” with the normalized *Artemisia* and xerophyte stacks. Therefore, the “Methods” section has been modified, including the new “Supplementary Table 2” to better understand the methodology used in the study, especially the part related with the pollen normalization and the stacked record (*Artemisia* and xerophytes). With respect to the raw pollen data, as explained above, they can be found in the open access Neotoma database (www.neotomadb.org) and European Pollen Database (www.europeanpollendatabase.net), whereas the private pollen records [seven pollen records: four records from other authors (Viñuelas, Brezoso, Medina, Zoñar) and three records studied by us (Padul, Rio Seco, Hondera)] can be found by contacting the authors of the original studies. We think that including the privately provided raw pollen data would go against the data privacy of the authors.

The comparison between the pollen record and historical is quite convincing, but the quality of the manuscript could be greatly improved by a longer and more detailed discussion on the specific socioeconomic impacts of droughts on the Visigoths, the caliphate and Emirate of Cordoba. I would suggest to delete the paragraph of the potential causes of the droughts to have more space available to move beyond the rather simplistic relationship between drought and collapse/change.

Thanks for the comment. However, there are very few historical sources about the socioeconomic and political aspects and impacts during the studied drought periods, not allowing us to offer a complete and detailed interpretation on this topic. Therefore, we believe

that further discussion on this matter could generate an overinterpretation of the real social, economic and political situation during those times.

Overall, the manuscript is very interesting and potentially suited for Nature Communications after major revisions were made. I would be willing to review the revised version of this manuscript.

Thank you for the positive feedback and for the constructive and helpful comments.

Reviewer #3 (Remarks to the Author):

REVIEW OF THE MANUSCRIPT ID NCOMMS-22-45602-T: 'Droughts contributed to the Visigothic Kingdom crisis and Islamic expansion in the Iberian Peninsula (7th to 10th centuries)' by Camuera et al.

This paper constitutes a very interesting contribution to Paleoclimatology with regional to global implications and provides new evidence on the potential influence of climate crisis on historical changes. In particular, the authors aim to demonstrate the influence of intense drought episodes on the Islamic expansion in the Iberian Peninsula and internal socioeconomic crises in Iberian Muslim kingdoms during the subsequent centuries. The climatic context of these historical changes has been relatively understudied compared to other relevant cultural changes, making this research timely and novel. Furthermore, this study integrates existing proxy data from pollen records of the Iberian Peninsula and Morocco into a high resolution hydroclimatic reconstruction able to register short-lived droughts during the last millennia, following an innovative methodological approach.

Therefore, this work deserves publication in the journal 'Nature Communications.' However, there are several issues that need to be addressed. Most of the points raised in the comments are of moderate importance and do not affect the main statements of the paper.

Thank you for the positive comment and for appreciating the work done. We believe that the changes based on your suggestions have greatly improved the new version of the article.

Main points:

1) The use of Artemisia pollen percentages as direct evidence of arid conditions remains controversial for recent chronologies (late Holocene) as this proxy has been also related with anthropogenic disturbances. In the Methods section (lines 252-256), the authors indicate that, according to the modern pollen database, Artemisia pollen percentages occur in percentages over 7% in regions below 400 mm/yr. However, the sensitivity of this proxy in the record of short-lived (decadal and subdecadal timescales) drought episodes needs further discussion. A potential correlation with other independent (i.e., non-pollen based) hydroclimatic reconstructions of comparable temporal resolution, such as recent instrumental and available tree ring records in the Iberian Peninsula and Morocco (Esper et al., 2007) and nearby regions for overlapping chronologies might contribute to reinforce the validity of this proxy.

Thanks for the comment. In the "Methods" section, with the calibration of "Artemisia vs annual precipitation" using the new Eurasian Modern Pollen Database of this new version of the manuscript, we demonstrate that *Artemisia* is not only one of the most important taxa related with arid conditions in the Mediterranean region in the past, but it also clearly responds to low precipitation conditions today. In our study we show that, at decadal timescales, extreme

drought periods cause an expansion of fast-growing and arid-adapted herb/grass species, such as *Artemisia*, resulting in higher proportion of these taxa in the fossil record.

We agree with the reviewer about the comparison of our pollen stack record with other paleoclimatic records from the area. In this respect, in this new version we have included a decadal-resolution stacked $\delta^{13}\text{C}$ and composite $\delta^{18}\text{O}$ speleothem records from northern Spain and northern Morocco (Martín-Chivelet et al., 2011; Ait Brahim et al., 2019). The high-resolution $\delta^{13}\text{C}$ and $\delta^{18}\text{O}$ records provide us with information about the vegetation density, CO_2 degassing of dripwater, soil biogenic CO_2 production via microbial activity and plant respiration, and/or the amount of precipitation, which are controlled by the paleohydrological conditions of the region. The comparison of our pollen data (stacked *Artemisia* record) with these isotopic records provides a new vision and interpretations of the paleohydrological conditions in the study region. By including these records, we have been able to cover the entire region from the northern Iberian latitude to Morocco, and therefore, obtain a robust paleoclimate information of this area for our period of interest.

With respect to the tree ring records from Iberia and Morocco, as far as we know, there are no records covering periods older than 1000 years BP. In fact, the suggested article from Esper et al. (2007) covers the last 953 years, not reaching our period of interest between 650 and 950 CE (1300-1000 cal yr BP).

2) The structure of the discussion on the main three droughts reconstructed in this research is unbalanced in the manuscript. A well-documented, detailed comparison with other regional and global palaeohydrological records is made for the first crisis (695-725 CE). In contrast, the last two episodes are exclusively discussed in terms of socio-economic consequences and historical background. The paleoclimatic context of the last two episodes should be also discussed with a similar degree of detail. As an alternative, the two paragraphs between lines 80 and 109 might be moved to the end of the paper, after the historical description of the three aridity crises, incorporating a comparison of obtained results with other paleoclimatic records of the region for the three aridity crises described in this paper.

We agree with the reviewer and in this new version we have modified the order of the paragraphs. The comparison of the *Artemisia* and xerophyte stacks with other Iberian and Moroccan paleohydrological records has been moved to the section before the historical description of the drought periods. In contrast to what the reviewer suggested, we believe that it is better to describe and discuss the paleoclimate records and their similarities/differences (including our *Artemisia*/xerophyte results, $\delta^{13}\text{C}$ and $\delta^{18}\text{O}$ records from the selected Iberian/Moroccan caves, the quantitative precipitation reconstructions from northern and southern Iberia, etc.) before the description and comparison with historical sources. However, we are open to modify the order of the sections if necessary.

*3) The *Artemisia* and xerophyte stacks presented in this paper are based on the whole dataset of available Iberian and NW African records. However, in the discussion (lines 78-94), these records are only correlated here with Southern Iberian paleoclimate records (Medina and Zoñar lake sediment sequences). A comparison with the precipitation reconstruction for Northern Iberia (available from Ilvonen et al. (2022)) and other terrestrial records is also required here to understand the consistency between pollen dataset and these reconstructions, at the same spatial scale. If drought episodes reconstructed in this research have been recorded (or not) with variable intensities in Northern Iberia, it should be also discussed here. In addition, in lines 95-109, the comparison of reconstructed droughts with the rainfall reconstruction for the Western*

Mediterranean region based on speleothem records from Morocco carried out by Ait Brahim et al. (2019) is relevant here and it should be also included in Extended Data Fig. 1 and discussed here. A potential comparison with the Buraca Gloriosa speleothem record (Western Portugal) (Thatcher et al., 2020), covering a shorter period (i.e., the last ca. 1200 years) might be also useful for the last drought discussed in this manuscript. In contrast, the correlation with Eastern Mediterranean records should be taken with care as significant differences in the hydrological evolution of Western and Eastern areas of the Mediterranean basin during the last millennium have been reconstructed at these timescales in relation with NAO forcing (Roberts et al., 2012). Although the aridity crisis reconstructed in this research are previous to 900 CE, these potential hydrological contrasts should be taken into account in the discussion (lines 108-109).

Thank you for the suggestion. We agree, and in this new version of the manuscript we compared our stacked *Artemisia* and xerophyte data with additional paleohydrological records. In fact, we have modified the section “Drought periods reconstructed from Iberian/Moroccan paleoenvironmental proxies” comparing our pollen data with new paleoclimate records, as the reviewers suggested. We have selected some of the suggested records and other new records, covering an area from northern Iberian latitudes to Morocco. In the new Figure 2 and Extended Data Figure 1 (Supplementary Figure 1) our stacked *Artemisia* and xerophyte data have been compared and correlated with other new paleoclimate proxy records, including: **(1)** the northern Iberian composite $\delta^{13}\text{C}$ record based on three speleothems from Cobre, Kaite and Mayor caves (N Spain); **(2)** composite $\delta^{18}\text{O}$ records based on two speleothems from Chaara cave (N Morocco); **(3)** composite $\delta^{18}\text{O}$ record from Buraca Gloriosa (Portugal); **(4)** not only southern Iberian reconstruction (as in the previous version of the manuscript) but also northern Iberian quantitative precipitation reconstructions from Ilvonen et al. (2022); **(5)** a new NAO index reconstruction for the Iberian Peninsula (apart from the previous NAO reconstruction included in the previous version of the manuscript). In addition to these new records, we have also maintained the $\delta^{18}\text{O}$ record from Asiul cave (N Spain), the *Artemisia* and *Gelasinospora* records from Medina Lake and the lake level reconstruction from Zoñar Lake.

4) The description of potential climatic mechanisms driving reconstructed aridity crises in the Iberian Peninsula are not sufficiently clear and lack detail. The relationship between arid and humid phases at centennial scale in the Western Mediterranean Region (e.g., Medieval Climate Anomaly and Little Ice Age) with NAO variability has been already discussed and evidenced in a number of paleoclimate works. However, what is the specific mechanism for these short-lived aridity crises occurring at decadal timescales? A more specific discussion on these potential mechanisms is required here.

Although we understand the reviewer's concern on this topic, the climatic mechanisms driving these short-lived drought periods identified and described in the paper are not the main goal of the manuscript. Even at present, how the NAO is affected by radiative forcing and the interdecadal variability of these modes remain important and unresolved questions in paleoclimate research.

Minor points:

Lines (L) 25-27) This sentence is unclear. I understand that, according to the authors, persistent droughts occurred during this period might have contributed to the destabilization of Muslim kingdoms ruling the Iberian Peninsula. Therefore, ‘internal rebellions and civil wars’ were a consequence of climate conditions and I suggest replacing ‘due to’ with ‘promoting’.

Changed.

L 34-36) The chronology of Islamic expansion in the circum-Mediterranean area is needed here.

We have included the chronology of Islamic expansion in the region.

L 36-38) The end of Visigothic Kingdom is repeated here and this sentence is a bit unclear. Please, replace 'until its fall during the early 8th century when' with 'the beginning of Islamic expansion in Iberia.'

We agree. We have changed it.

L39) I think '(Islamic dynasty)' might be deleted here.

We think that it might be useful for an unexperienced reader to specify this.

L42) Please, replace 'have' by 'has'

Done.

L42-43) Although the number of archaeological studies of Visigothic settlements is comparatively lower than others in the region, this sentence is too conclusive. Please, rephrase.

Since the number of archaeological records and studies/analyses carried out for the Visigothic period are lower than for other periods (for example, for the period of the Roman Empire), we think that the sentence captures well what we want to express.

L45-46) There is a considerable number of terrestrial, high-resolution records in Spain covering the Middle Ages, compared to other periods (e.g., late glacial). In fact, the authors cite most of these works and have used pollen data for this research. Please, rephrase.

We have modified the sentence. However, although we have used pollen records for this study, the resolution of these records individually is low to reach the conclusions obtained in this study.

L74) 'Is' instead of 'are'.

Changed. Note that this sentence has been moved to another part of the manuscript due to the modification of the sections, as we have separated the paleoclimatic records and interpretations from the sections about historical sources.

L194) A new, specific North Atlantic Reconstruction (NAO) for the Iberian Peninsula is available (Hernández et al., 2020) and it is probably more adequate for its comparison with reconstructed hydroclimatic conditions and the discussion of potential driving mechanisms for the investigated aridity crisis. I also suggest incorporating this reconstruction to Fig. 2 and Extended Data Fig. 1.

Thanks for the suggestion. As we have explained above, we have included the Iberian NAO reconstruction. Although this NAO reconstruction for the Iberian Peninsula does not perfectly match our pollen data, we have included it in Extended Data Figure 1 and briefly discussed it in the text (Lines 215-219).

REFERENCES

Ait Brahim, Y., Wassenburg, J. A., Sha, L., Cruz, F. W., Deininger, M., Sifeddine, A., . . . Cheng, H. (2019). North Atlantic Ice-Rafting, Ocean and Atmospheric Circulation During the Holocene: Insights From Western Mediterranean Speleothems. *Geophysical Research Letters*, 46(13), 7614-7623. doi:<https://doi.org/10.1029/2019GL082405>

Esper, J., Frank, D., Büntgen, U., Verstege, A., Luterbacher, J., & Xoplaki, E. (2007). Long-term drought severity variations in Morocco. *Geophys. Res. Lett.*, 34(17), L17702. Retrieved from <http://dx.doi.org/10.1029/2007GL030844>

Hernández, A., Sánchez-López, G., Pla-Rabes, S., Comas-Bru, L., Parnell, A., Cahill, N., . . . Giralt, S. (2020). A 2,000-year Bayesian NAO reconstruction from the Iberian Peninsula. *Scientific Reports*, 10(1), 14961. doi:10.1038/s41598-020-71372-5

Ilvonen, L., López-Sáez, J. A., Holmström, L., Alba-Sánchez, F., Pérez-Díaz, S., Carrión, J. S., . . . Seppä, H. (2022). Spatial and temporal patterns of Holocene precipitation change in the Iberian Peninsula. *Boreas*, 51(4), 776-792. doi:<https://doi.org/10.1111/bor.12586>

Roberts, N., Moreno, A., Valero-Garcés, B. L., Corella, J. P., Jones, M., Allcock, S., . . . Türkeş, M. (2012). Palaeolimnological evidence for an east–west climate see-saw in the Mediterranean since AD 900. *Global and Planetary Change*, 84-85, 23-24. doi:10.1016/j.gloplacha.2011.11.002

Thatcher, D. L., Wanamaker, A. D., Denniston, R. F., Asmerom, Y., Polyak, V. J., Fullick, D., . . . Haws, J. A. (2020). Hydroclimate variability from western Iberia (Portugal) during the Holocene: Insights from a composite stalagmite isotope record. *The Holocene*, 30(7), 966-981. doi:10.1177/0959683620908648

REFERENCES

Ait Brahim, Y., Wassenburg, J.A., Sha, L., Cruz, F.W., Deininger, M., Sifeddine, A., Bouchaou, L., Spötl, C., Edwards, R.L., Cheng, H., 2019. North Atlantic Ice-Rafting, Ocean and Atmospheric Circulation During the Holocene: Insights From Western Mediterranean Speleothems. *Geophysical Research Letters* 46, 7614-7623. <https://doi.org/10.1029/2019GL082405>.

Carrión, J.S., Fernández, S., González-Sampériz, P., Gil-Romera, G., Badal, E., Carrión-Marco, Y., López-Merino, L., López-Sáez, J.A., Fierro, E., Burjachs, F., 2010. Expected trends and surprises in the Lateglacial and Holocene vegetation history of the Iberian Peninsula and Balearic Islands. *Review of Palaeobotany and Palynology* 162, 458-475. <https://doi.org/10.1016/j.revpalbo.2009.12.007>.

Fleitmann, D., Haldon, J., Bradley, R.S., Burns, S.J., Cheng, H., Edwards, R.L., Raible, C.C., Jacobson, M., Matter, A., 2022. Droughts and societal change: The environmental context for the emergence of Islam in late Antique Arabia. *Science* 376, 1317-1321. [10.1126/science.abg4044](https://doi.org/10.1126/science.abg4044).

Ilvonen, L., López-Sáez, J.A., Holmström, L., Alba-Sánchez, F., Pérez-Díaz, S., Carrión, J.S., Ramos-Román, M.J., Camuera, J., Jiménez-Moreno, G., Ruha, L., Seppä, H., 2022. Spatial and temporal patterns of Holocene precipitation change in the Iberian Peninsula. *Boreas* 51, 776-792. <https://doi.org/10.1111/bor.12586>.

Jiménez-Moreno, G., Anderson, R.S., Ramos-Román, M.J., Camuera, J., Mesa-Fernández, J.M., García-Alix, A., Jiménez-Espejo, F.J., Carrión, J.S., López-Avilés, A., 2020. The Holocene Cedrus pollen record from Sierra Nevada (S Spain), a proxy for climate change in N Africa. *Quaternary Science Reviews* 242, 106468. <https://doi.org/10.1016/j.quascirev.2020.106468>.

Martín-Chivelet, J., Muñoz-García, M.B., Edwards, R.L., Turrero, M.J., Ortega, A.I., 2011. Land surface temperature changes in Northern Iberia since 4000yrBP, based on $\delta^{13}\text{C}$ of speleothems. *Global and Planetary Change* 77, 1-12. <https://doi.org/10.1016/j.gloplacha.2011.02.002>.

Olmo-Enciso, L., Castro-Priego, M., Ruiz Zapata, B., Gil García, M., Galindo Pellicena, M., Checa-Herráiz, J., Gómez de la Torre-Verdejo, A.J.M.L.i.P.A.N.F., Perspectives, N., 2019. The Construction and Dynamics of Early Medieval Landscapes in Central Iberia, In: Gelichi, S., Olmo-

Enciso, L. (Eds.), Mediterranean Landscapes in Post Antiquity - New frontiers and new perspectives. Archaeopress Publishing Ltd, pp. 104-128.

Pini, R., Furlanetto, G., Vallé, F., Badino, F., Wick, L., Anselmetti, F.S., Bertuletti, P., Fusi, N., Morlock, M.A., Delmonte, B., Harrison, S.P., Maggi, V., Ravazzi, C., 2022. Linking North Atlantic and Alpine Last Glacial Maximum climates via a high-resolution pollen-based subarctic forest steppe record. *Quaternary Science Reviews* 294, 107759. <https://doi.org/10.1016/j.quascirev.2022.107759>.

Thatcher, D.L., Wanamaker, A.D., Denniston, R.F., Asmerom, Y., Polyak, V.J., Fullick, D., Ummenhofer, C.C., Gillikin, D.P., Haws, J.A.J.T.H., 2020. Hydroclimate variability from western Iberia (Portugal) during the Holocene: Insights from a composite stalagmite isotope record. *The Holocene* 30, 966-981.

Reviewers' Comments:

Reviewer #1:

Remarks to the Author:

The revised version of the paper has consistently been improved. Mostly, it now nuances the direct (even deterministic) relationship between droughts and historical events from the prior version. However, I still disagree with the interpretation of results for the same reasons I mentioned in my review and I agree with reviewer 2 when she/he writes that the paper needs "to move beyond the rather simplistic relationship between drought and collapse/change". In summary, it does not make clear how exactly droughts made a state collapse beyond saying that the Visigothic state was "weak" and establishing rather arbitrary connections between historical events (mainly through documentary analysis and occasionally using archaeological record, mainly the city of Reccopolis). I insist that this is complex enough to be solved in just 8 pages.

Having said this, I recommend its publication due to the evident effort of nuancing the strongest claim of a direct relationship between climate and historical crises. Even though I disagree with the interpretation and still think it deserves a more thorough and complete analysis, this is not a good reason for denying the possibility to put it in open discussion with other colleagues.

Just an additional point. I disagree with the comment that isotope analysis cannot identify diet problems and stress (see Beaumont and Montgomery, 2016 in the bibliography). For instance, if the paper's hypothesis is true, it should be expected some dietary changes throughout the period, something that is not seen in the archaeological record or at least not discussed by the authors. I include some references (as the authors asked in their response) to support my concerns with the paper, considering the debates on the collapse of the visigothic state and the emergence of andalusian one as well as the isotope question:

Beaumont, J., y Montgomery, J. (2016) "The great Irish famine: identifying starvation in the tissues of victims using stable isotope analysis of bone and incremental dentine collagen", *PLoS One*, 11, 8.

Carvajal Castro, Á., y Tejerizo García, C. (2023) *El Estado y la Alta Edad Media. Nuevas perspectivas*. Bilbao, Universidad del País Vasco.

García-Collado, M. I. (2016) "Food consumption patterns and social inequality in an early medieval rural community in the centre of the Iberian Peninsula", en J. A. Quirós Castillo (Ed.), *Social complexity in Early Medieval Rural Communities. The north-western Iberia Archaeological Record*, Oxford, Archaeopress, pp. 59-78.

Manzano Moreno, E. (2023) "Aspectos políticos ideológicos en la configuración del Estado omeya en al-Andalus", en Á. Carvajal Castro y C. Tejerizo García (Eds.), *El Estado y la Alta Edad Media. Nuevas perspectivas*, Bilbao, Universidad del País Vasco, pp. 29-42.

Martín Viso, I. (2021) "Tiempos de colapso y resiliencia: espacios sin estado en la Península Ibérica (siglos VIII-X)", *Intus-Legere Historia*, 15, 2, pp. 78-105.

Poveda, P. (2018) *Monarquía, sociedad y territorio en el Occidente post-imperial: construcción de la soberanía regia en la Galia merovingia y en el reino visigodo hispano*. Tesis doctoral inédita presentada en la Universidad de Salamanca.

Quirós Castillo, J. A., Loza Uriarte, M., y Niso Lorenzo, J. (2013) "Identidades y ajueres en las necrópolis altomedievales. Estudios isotópicos del cementerio de San Martín de Dulantzi, Álava (siglos VI-X)", *Archivo Español de Arqueología*, 86, pp. 215-232.

Vigil-Escalera Guirado, A. (2015) "La identidad de la comunidad local y las afiliaciones individuales en necrópolis de la Alta Edad Media", en J. A. Quirós Castillo y S. Castellanos (Eds.), *Identidad y etnicidad en Hispania. Propuestas teóricas y cultura material en los siglos V-VIII*, Bilbao, Universidad del País Vasco, pp. 249-274.

Reviewer #3:

Remarks to the Author:

REVIEW OF THE MANUSCRIPT ID NCOMMS-22-45602B: 'Droughts contributed to the Visigothic Kingdom crisis and Islamic expansion in the Iberian Peninsula (5th to 10th centuries CE)' by Camuera

et al.

The authors have provided satisfactory answers and detailed explanations to the main points raised in my previous comments. Moreover, the quality of the manuscript has improved significantly after the modifications carried out by the authors, particularly on its structure and figures. Thus, I recommend the acceptance of this article on its present form in the journal 'Nature Communications.' There are still a couple of minor suggestions on the revised version of the manuscript that should be considered: Lines 191 to 193 (on the revised version): According to the authors, the impact of recorded drought episodes in the Iberian Peninsula over larger areas of the Western Mediterranean, particularly Northern Africa, might have favored migrations and expansion of Islamic people towards northern latitudes (subjected to moister conditions). This statement makes sense but needs to be supported on reference(s) demonstrating the occurrence of arid conditions in North Africa during the 8th century. This hypothesis has already been used to explain migrations in late Antique Arabia -in the context of the emergence of Islam- during the 7th century CE (Fleitmann et al., 2019).

L217-220) The same applies to this sentence, which needs to be supported by reference(s).

REFERENCES CITED IN THIS REVIEW

Fleitmann, D., Haldon, J., Bradley, R. S., Burns, S. J., Cheng, H., Edwards, R. L., . . . Matter, A. (2022). Droughts and societal change: The environmental context for the emergence of Islam in late Antique Arabia. *Science*, 376(6599), 1317-1321. doi:doi:10.1126/science.abg4044

Reviewer #4:

Remarks to the Author:

Climate variations in the historical sequence have gained prominence in recent research and it is commonly accepted that climate change has an influential effect on human societies. However, there is no unanimous opinion on the extent to which climate influences episodes of prosperity or crisis of nations and empires. Some of the interpretations that have been made of the impact of climate on the crises of states are deterministic, to the point of proposing that the demise of the Roman Empire could be directly related to the droughts that affected the Mediterranean region from the 4th century onwards, the beginning of the Dark Ages Cold Period, DACP, 400-700 (Harper, 2017). Some authors have suggested that the survival of the Byzantine Empire might have been favoured by a lower incidence of unfavourable climatic changes in the eastern part of the Mediterranean (McCormick et al., 2012: 197). It has also been suggested that the crisis affecting rural settlements in parts of Europe between the 5th and 7th centuries could be related to the cold and dry episode known as the DACP (Cheyette, 2008). This deterministic position is not accepted by other researchers who suggest that attention should be directed towards the adaptive responses of human communities to climatic variations, which can be highly variable (Jalut et al., 2009; Walsh, 2005).

The thesis of the authors of the article under review seems to be closer to the deterministic reading than to the more nuanced interpretations (this is something that has also been pointed out by some of the other reviewers). The authors have responded to this criticism by arguing that their proposal is to consider droughts as one of the agents that caused the rise and fall of the early medieval kingdoms of the Iberian Peninsula, but not the only one. However, from the reading of their text, it seems to have been concluded that the climatic crisis was a determining factor in the end of the Visigothic kingdom (in fact, this seems to be the main thesis of the contribution). This interpretation fails to explain why droughts would have been an influential factor in the demise of the Visigothic kingdom but would have had little effect on the Umayyad emirate, which replaced it with complete success. Historians generally attribute to the Umayyad emirate and caliphate high levels of urban development, efficient exploitation of agricultural areas and a very active trade network (Manzano, 2006), phenomena that would have taken place under the same climatic conditions that supposedly contributed to the ruin of the Visigothic kingdom. There is no reason to suppose that the economic bases of these two states

were very different and both probably exploited the available resources in a similar way.

Without questioning the value of pollen records for the study of climatic changes in the historical sequence, it is necessary to recall that the vegetation landscape is also affected by the strategies employed by human populations to cope with such changes. In the Central System, important deforestation processes associated with the Visigothic period have been detected, related to climatic changes, and significant anthropic pressure aimed at transforming mountain areas into pastureland. The historical sequence corresponding to the period of Islamic domination is characterised in the same region by a certain forest recovery, a phenomenon that occurs under very similar climatic conditions (López-Sáez et al., 2014).

It must be noted that the study is highly dependent on pollen data, although there are other records that provide information on the climate of the period under analysis. The authors are invited to consider whether to include some references to these studies in the article. I refer especially to the analysis of variations in mercury concentrations in the Penido Vello deposits (Martínez-Cortizas et al., 1999), as well as to the investigation of some marine sediments. Specifically, data obtained in marine deposits from the Tagus estuary (Abrantes et al., 2005; Rodrigues et al., 2009), from the Vigo estuary (Álvarez et al., 2005) and from the western Mediterranean basin (Nieto-Moreno et al., 2011, cited by the authors) are relevant.

The presentation of archaeological data is very brief (it is accepted that this is not the aim of the work). However, the contribution by Martín-Viso used as the main reference (Martín Viso, 2013) deals with a specific problem of regional scope (the archaeological contexts of the 'Visigothic slates'). This same author has published another study much more closely related to the problem to be addressed (Martín Viso, 2012). On the other hand, it is not true that archaeological data referring to the Visigothic period are scarce. An update of the very abundant archaeological information on the period can be found in a recent monograph (Diarte-Blasco, 2018). On the archaeology of Islam in the Iberian Peninsula, there are also syntheses to be considered (Manzano, 2006: 238-216).

It is not entirely certain that the Visigothic kingdom faced a serious political and administrative crisis before the Islamic conquest. This traditional reading is being challenged by more recent research, which has questioned the decadence which has been attributed to it in the 7th century and emphasised instead the dynamic and reformist character of the Visigothic state in the years before its demise (Díaz, 2012).

The contribution is judged to be consistent in its presentation of the data (although improvements are suggested). The fact that I do not share the authors' point of view on the determinant character of climatic changes in the crisis of the early medieval kingdoms should not be considered an obstacle to its publication.

In lines 161-162: The Visigoths tried to adapt to such drought conditions in the 7th and 8th centuries CE using different techniques. The fact is that the Visigothic kingdom disappeared in 711, so this chronological framework needs to be better defined.

References

Abrantes, F., Lebreiro, S., Rodrigues, T., Gil, I., Bartels-Jónsdóttir, H., Oliveira, P., Kissel, C & Grimalt, J.O. Shallow-marine sediment cores record climate variability and earthquake activity off Lisbon (Portugal) for the last 2000 years. *Quaternary Science Reviews* 24, 2477-3494 (2005).

Álvarez, M.C., Flores, J.A., Siirro, F.J., Diz, P., Francés, G., Pelejero, C. & Grimalt, J. Millennial surface water dynamics in the Ría de Vigo during the last 3000 years as revealed by coccoliths and molecular biomarkers. *Palaeogeography, Palaeoclimatology, Palaeoecology* 218, 1-13 (2005).

Cheyette, F.L. The disappearance of the ancient landscape and the climatic anomaly of the early Middle Ages: a question to be pursued. *Early Medieval Europe* 16(2), 127-165 (2008).

Diarte-Blasco, P. Late Antique and Early Medieval Hispania. Landscapes without Strategy. An archaeological approach (Oxbow Books, 2018).

Díaz, P.C. La dinámica del poder y la defensa del territorio: para una comprensión del fin del reino visigodo de Toledo, In *Actas de la XXXIX semana de estudios medievales. |De Mahoma a Carlomagno. Los primeros tiempos (siglos VII-IX)*, 168-205 (2012).

Harper, K. *The Fate of Rome. Climate, Disease and the End of an Empire* (Princeton University Press, 2017).

Jalut, G., Dedoubat, J.-J., Fontugne, M., Otto, T. Holocene circum-Mediterranean vegetation changes: Climate forcing and human impact. *Quaternary International* 200, 4-18 (2009).

López-Sáez, J.A., Abel-Schaad, D., Pérez-Díaz, S., Blanco-González, A., Alba-Sánchez, F., Dorado, M., Ruiz-Zapata, B., Gil-García, M.J., Gómez-González, C. & Franco-Música, F. Vegetation history, climate and human impact in the Spanish Central System over the last 9000 years. *Quaternary International* 353, 98-122 (2014).

Manzano, E. *Conquistadores, emires y califas. Los Omeyas y la formación de al-Andalus* (Crítica, 2006).

Martín Viso, I. The "Visigothic" slates and their archaeological contexts. *Journal of 593 Medieval Iberian Studies* 5, 145-168 (2013).

Martín Viso, I. Un mundo en transformación: los espacios rurales en la Hispania post-romana (siglos V-VII). In *Visigodos y Omeyas. El territorio* (ed L. Caballero, P. Mateos & T. Cordero) (Consejo Superior de Investigaciones Científicas, 2012) 31-63.

Martínez-Cortizas, A., Pontevedra-Pombal, X., García-Rodeja, E., Nóvoa-Muñoz, J.C. & Shotyk, W. Mercury in a Spanish peat bog: archive of climate change and atmospheric metal deposition. *Science* 284, 939-942 (1999).

McCormick, M., Büntgen, U., Cane, M.A., Cook, E.R., Harper, K., Huybers, P., Litt, T., Manning, S.W., Mayewski, P.A. More, A.F.M. Nicolussi, K. & Tegel, W. Climate Change during and after the Roman Empire: Reconstructing the Past from Scientific and Historical Evidence. *Journal of Interdisciplinary History* 43(2), 169-220 (2012).

Nieto-Moreno, V., Martínez-Ruiz, F., Giral, S., Jiménez-Espejo, F., Gallego-Torres, D., Rodrigo-Gámiz, M., García-Orellana, J., Ortega-Huertas, M. & de Lange, J.G. Tracking climate variability in the western Mediterranean during the Late Holocene: a multiproxy approach. *Climate of the Past* 7, 1395-1414 (2011).

Rodrigues, T., Grimalt, J.O., Abrantes, F.G., Flores, J.A. and Lebreiro, S.M. Holocene interdependences of changes in sea surface temperature, productivity, and fluvial inputs in the Iberian continental shelf (Tagus mud patch). *Geochemistry, Geophysics, Geosystems* 10(7), 1-17 (2009).

Walsh, K. Risk and marginality at high altitudes: new interpretations from fieldwork on the Faravel Plateau, Hautes-Alpes. *Antiquity* 79, 289-305 (2005).

The reviewers' comments are in **black** and our responses in **blue**.

In this file, when we reference lines (e.g., "Lines 226-229"), we refer to the lines of the clean manuscript file "*Manuscript Camuera et al_NCOMMS-22-45602B_July2023_MainText*", which does not have the *Track Changes* activated. We have also uploaded the file with *Track changes* activated, which includes all changes performed in this revision (file "*TrackChanges_Manuscript Camuera et al_NCOMMS-22-45602B_July2023_MainText*"). In addition, the Supplementary Figures 1 to 3 and Supplementary Table 1 have been included separately in the new Supplementary Information file. The modified version of the Supplementary Table 1 has also been uploaded to the Manuscript Tracking System.

Reviewer #1 (Remarks to the Author):

The revised version of the paper has consistently been improved. Mostly, it now nuances the direct (even deterministic) relationship between droughts and historical events from the prior version. However, I still disagree with the interpretation of results for the same reasons I mentioned in my review and I agree with reviewer 2 when she/he writes that the paper needs "to move beyond the rather simplistic relationship between drought and collapse/change". In summary, it does not make clear how exactly droughts made a state collapse beyond saying that the Visigothic state was "weak" and establishing rather arbitrary connections between historical events (mainly through documentary analysis and occasionally using archaeological record, mainly the city of Reccopolis). I insist that this is complex enough to be solved in just 8 pages.

We are grateful for the reviewer's comments. We have followed the reviewer's suggestions and have tried to reformulate some of the concepts used.

We agree with the Reviewer #1 when they say that this is complex enough to be solved in a short manuscript. However, as we mentioned in previous revisions, this study focuses on the climate conditions occurring between 5th and 10th centuries CE, and we have attempted to give appropriate and contrasted interpretations of the responses of the kingdoms to these climate conditions/changes that occurred during periods of huge historical relevance in few pages. However, despite the temporal coincidences between the droughts recorded by the vegetation and by the historical sources, we have included a series of observations in the text so that a lesser weight of a deterministic interpretation can be now observed in the climate analysis of the studied period and the historical events.

We have answered all these raised archaeological questions with the help of a new co-author specialized in the archaeology of the Visigothic period and, taking into account the limited space for the explanation of these concepts and on the basis of a critical review, some observations have been added and clarified. The Early Medieval archaeology and a new reinterpretation of the written sources have generated new revulsive studies on different aspects surrounding the Visigothic period. Up-to-date bibliographical information about the subject can be found in *Archaeology and history of peasantries* (Quirós-Castillo, 2020). In results and discussion, we have pointed out how one of the central elements of research in recent decades has been the characterization of peasant spaces during Early Medieval Iberia. The identification of these spaces has allowed a new interpretation of the strategies of resource capture. Undoubtedly, one of the most interesting topics is the expansion of livestock farming and its relationship with open landscapes. We have also included new data and references, which show the capacity of peasant

communities to adapt to a changing environmental and climate conditions, with strategies far removed from the dynamics imposed by the states (Lines 185-198).

Regarding the comment about "going beyond the rather simplistic relationship between drought and collapse/change", we have tried to give it a deeper interpretation. For example, in this regard, it is worth mentioning that some authors noted that the presence of the Muslim rule in Iberia and in other Mediterranean regions during the Islamic expansion from the Middle-East to Iberia (7th – 8th centuries CE) could have also been favoured by the introduction and spread of new technology and agricultural practices and products, the so-called Green Revolution (Watson 1974). However, there is increasing evidence for the diffusion of pre-Islamic agricultural techniques, tools and crops adapted to semi-arid regions, suggesting that the pre- and post-Islamic Near East and Mediterranean areas were more similar to each other than previously recognized (Decker 2009). We have included both interpretations in the text (Lines 230-255). However, as mentioned above and in the new version of the manuscript, there is no consensus in the scientific community on this matter and it is open to debate.

Having said this, I recommend its publication due to the evident effort of nuancing the strongest claim of a direct relationship between climate and historical crises. Even though I disagree with the interpretation and still think it deserves a more thorough and complete analysis, this is not a good reason for denying the possibility to put it in open discussion with other colleagues.

Thanks for the comment and the constructive feedback.

Just an additional point. I disagree with the comment that isotope analysis cannot identify diet problems and stress (see Beaumont and Montgomery, 2016 in the bibliography). For instance, if the paper's hypothesis is true, it should be expected some dietary changes throughout the period, something that is not seen in the archaeological record or at least not discussed by the authors.

Thank you for the reference. We didn't know about this study related with the Great Famine of the 19th century CE. We have revised the bibliography about isotope changes ($\delta^{15}\text{N}$, $\delta^{13}\text{C}$) analyzed in single individuals in Iberia for the 5th – 10th centuries CE, which could show changes in diet and/or stress during the life of the individual. Isotopic studies carried out in Iberia for this time period show changes in the diet of different populations over long time periods, such as differences between Visigothic, Muslim and Christian diets (Alexander et al., 2019), the changing diet between different Iberian regions from Medieval to Early Modern Spain (MacKinnon, 2015), and the diet diversity and the geographic origin/mobility of several individuals in Christian and Muslim areas between 8th and 10th centuries CE (Guede et al., 2017). However, as far as we know, there is a lack of isotopic studies about abrupt changes in diets, stress and/or famine in Iberia for our period of interest.

I include some references (as the authors asked in their response) to support my concerns with the paper, considering the debates on the collapse of the visigothic state and the emergence of andalusian one as well as the isotope question:

Beaumont, J., y Montgomery, J. (2016) "The great Irish famine: identifying starvation in the tissues of victims using stable isotope analysis of bone and incremental dentine collagen", PLoS One, 11, 8.

Carvajal Castro, Á., y Tejerizo García, C. (2023) El Estado y la Alta Edad Media. Nuevas perspectivas. Bilbao, Universidad del País Vasco.

García-Collado, M. I. (2016) "Food consumption patterns and social inequality in an early medieval rural community in the centre of the Iberian Peninsula", en J. A. Quirós Castillo (Ed.),

Social complexity in Early Medieval Rural Communities. The north-western Iberia Archaeological Record, Oxford, Archaeopress, pp. 59-78.

Manzano Moreno, E. (2023) "Aspectos políticos ideológicos en la configuración del Estado omeya en al-Andalus", en Á. Carvajal Castro y C. Tejerizo García (Eds.), *El Estado y la Alta Edad Media. Nuevas perspectivas*, Bilbao, Universidad del País Vasco, pp. 29-42.

Martín Viso, I. (2021) "Tiempos de colapso y resiliencia: espacios sin estado en la Península Ibérica (siglos VIII-X)", *Intus-Legere Historia*, 15, 2, pp. 78-105.

Poveda, P. (2018) *Monarquía, sociedad y territorio en el Occidente post-imperial: construcción de la soberanía regia en la Galia merovingia y en el reino visigodo hispano*. Tesis doctoral inédita presentada en la Universidad de Salamanca.

Quirós Castillo, J. A., Loza Uriarte, M., y Niso Lorenzo, J. (2013) "Identidades y ajuares en las necrópolis altomedievales. Estudios isotópicos del cementerio de San Martín de Dulantzi, Álava (siglos VI-X)", *Archivo Español de Arqueología*, 86, pp. 215-232.

Vigil-Escalera Guirado, A. (2015) "La identidad de la comunidad local y las afiliaciones individuales en necrópolis de la Alta Edad Media", en J. A. Quirós Castillo y S. Castellanos (Eds.), *Identidad y etnicidad en Hispania. Propuestas teóricas y cultura material en los siglos V-VIII*, Bilbao, Universidad del País Vasco, pp. 249-274.

References:

Castillo, J. A. Q. *Archaeology and History of Peasantries 2* (2020).

Watson, A. M. *The Arab agricultural revolution and its diffusion, 700–1100*. *The Journal of Economic History* 34, 8-35 (1974).

Decker, M. *Plants and progress: rethinking the Islamic agricultural revolution*. *Journal of World History* 20, 187-206 (2009).

Alexander, Michelle M., et al. *Economic and socio-cultural consequences of changing political rule on human and faunal diets in medieval Valencia (c. fifth–fifteenth century AD) as evidenced by stable isotopes*. *Archaeological and Anthropological Sciences* 11, 3875-3893 (2019).

MacKinnon, Amy T. *Dietary reconstruction of medieval and early modern Spanish populations using stable isotopes of carbon and nitrogen* (2015).

Guede, Irujo, et al. *Isotope analyses to explore diet and mobility in a medieval Muslim population at Tauste (NE Spain)*. *PLoS One* e0176572 (2017).

Reviewer #3 (Remarks to the Author):

REVIEW OF THE MANUSCRIPT ID NCOMMS-22-45602B: 'Droughts contributed to the Visigothic Kingdom crisis and Islamic expansion in the Iberian Peninsula (5th to 10th centuries CE)' by Camuera et al.

The authors have provided satisfactory answers and detailed explanations to the main points raised in my previous comments. Moreover, the quality of the manuscript has improved significantly after the modifications carried out by the authors, particularly on its structure and figures. Thus, I recommend the acceptance of this article on its present form in the journal 'Nature Communications.' There are still a couple of minor suggestions on the revised version of the manuscript that should be considered:

Lines 191 to 193 (on the revised version): According to the authors, the impact of recorded drought episodes in the Iberian Peninsula over larger areas of the Western Mediterranean, particularly Northern Africa, might have favored migrations and expansion of Islamic people towards northern latitudes (subjected to moister conditions). This statement makes sense but needs to be supported on reference(s) demonstrating the occurrence of arid conditions in North Africa during the 8th century. This hypothesis has already been used to explain migrations in late Antique Arabia -in the context of the emergence of Islam- during the 7th century CE (Fleitmann et al., 2019).

Thanks for the suggestion. Since there is no consensus in the scientific community about whether agricultural techniques, irrigation systems and hydraulic structures during the Muslim period allowed a better use of water or not with respect to the Visigothic period, we have not adopted a single interpretation for the "Results and discussion" section of the article (Lines 230-248). As the reviewer indicates, we have also included a new reference showing the Islamic expansion during the 6th century CE in the Arabian Peninsula related with arid conditions and strong drought periods, and the effect on the Islamic migration (Fleitmann et al., 2022) (Lines 236-240).

L217-220) The same applies to this sentence, which needs to be supported by reference(s).

Thank you for the suggestion. We have included a reference.

REFERENCES CITED IN THIS REVIEW

Fleitmann, D., Haldon, J., Bradley, R. S., Burns, S. J., Cheng, H., Edwards, R. L., ... Matter, A. (2022). Droughts and societal change: The environmental context for the emergence of Islam in late Antique Arabia. *Science*, 376(6599), 1317-1321. doi:10.1126/science.abg4044

Reviewer #4 (Remarks to the Author):

Climate variations in the historical sequence have gained prominence in recent research and it is commonly accepted that climate change has an influential effect on human societies. However, there is no unanimous opinion on the extent to which climate influences episodes of prosperity or crisis of nations and empires. Some of the interpretations that have been made of the impact of climate on the crises of states are deterministic, to the point of proposing that the demise of the Roman Empire could be directly related to the droughts that affected the Mediterranean region from the 4th century onwards, the beginning of the Dark Ages Cold Period, DACP, 400-700 (Harper, 2017). Some authors have suggested that the survival of the Byzantine Empire might have been favoured by a lower incidence of unfavourable climatic changes in the eastern part of the Mediterranean (McCormick et al., 2012: 197). It has also been suggested that the crisis affecting rural settlements in parts of Europe between the 5th and 7th centuries could be related to the cold and dry episode known as the DACP (Cheyette, 2008). This deterministic position is not accepted by other researchers who suggest that attention should be directed towards the adaptive responses of human communities to climatic variations, which can be highly variable (Jalut et al., 2009; Walsh, 2005).

Thank you for the comment. We have included a series of remarks in the text, so a lesser weight of a deterministic interpretation can be observed in the analysis of the climate of the period analysed. It is true that Early Medieval archaeology and new reinterpretations of the written sources in the Iberian Peninsula have been revulsive in the traditional questions about the Visigothic period. In particular, the identification of the social space of the peasantry has been the central element of debate during the last decades. Without any doubt, archaeology has shown two central elements: long-lasting social processes and the ability to adapt to environmental conditions that were certainly changing.

In an attempt to answer all these questions and based on the constructive reviews, some observations have been added or qualified, as we point out below. All of them are intended to make more comprehensible the interaction between the various disciplines that are providing new contributions to the long debate on the Visigothic and Paleo-Andalusian period.

The thesis of the authors of the article under review seems to be closer to the deterministic reading than to the more nuanced interpretations (this is something that has also been pointed out by some of the other reviewers). The authors have responded to this criticism by arguing that their proposal is to consider droughts as one of the agents that caused the rise and fall of the early medieval kingdoms of the Iberian Peninsula, but not the only one. However, from the reading of their text, it seems to have been concluded that the climatic crisis was a determining factor in the end of the Visigothic kingdom (in fact, this seems to be the main thesis of the contribution). This interpretation fails to explain why droughts would have been an influential factor in the demise of the Visigothic kingdom but would have had little effect on the Umayyad emirate, which replaced it with complete success. Historians generally attribute to the Umayyad emirate and caliphate high levels of urban development, efficient exploitation of agricultural areas and a very active trade network (Manzano, 2006), phenomena that would have taken place under the same climatic conditions that supposedly contributed to the ruin of the Visigothic kingdom. There is no reason to suppose that the economic bases of these two states were very different and both probably exploited the available resources in a similar way.

Thank you for the suggestion. As the reviewer mentioned, over the last decades different hypotheses have been suggested about the distribution of population and settlements in some parts of Europe in relation to climatic changes (such as during the DACP). We agree with the

reviewer in that we should not position ourselves in a climatically deterministic scenario, since there is not enough archaeological evidence for it. Therefore, we have softened the text suggesting the climatic crisis as a possible factor that could have contributed to the decline of the Visigothic kingdom as well as favoring the Islamic expansion in the Iberian Peninsula. Some authors indicated that that Islamic expansion may have been favoured by the introduction and diffusion of new technologies, and agricultural practices and products better adapted to drylands and arid environmental conditions (Watson, 1974). However, other studies have also shown that pre-Islamic agricultural techniques and the use of different agricultural products were not so different from those introduced by the Muslims in the Iberian Peninsula (Lines 242-248). Although from reading the article it can be deduced that the statistically significant extreme droughts could have affected the fall of the Visigothic kingdom (as well as the Visigothic civil war during the 6th CE century and the internal conflicts during the Islamic rule of Iberia – al-Andalus – in the 8th and 10th centuries CE), the main goal of the manuscript is to identify the arid conditions and strong droughts that occurred between the 5th and 10th centuries CE based on pollen records, and the possible coincidences with historical events deduced from historical sources and archaeological remains.

Without questioning the value of pollen records for the study of climatic changes in the historical sequence, it is necessary to recall that the vegetation landscape is also affected by the strategies employed by human populations to cope with such changes. In the Central System, important deforestation processes associated with the Visigothic period have been detected, related to climatic changes, and significant anthropic pressure aimed at transforming mountain areas into pastureland. The historical sequence corresponding to the period of Islamic domination is characterised in the same region by a certain forest recovery, a phenomenon that occurs under very similar climatic conditions (López-Sáez et al., 2014).

Thank you very much for the comment. The highly-cited article by López-Sáez et al. (2014) shows a very good regional view of the vegetation and, therefore, climatic and/or anthropogenic changes occurred during the last 9000 years mainly in the Spanish Central System. According to this paper, *“The Visigothic kingdom (cal AD 450-711) that evolved in Hispania after the Late Roman Empire was based on subsistence farming, and the fall of international trade eased the pressure on forest resources (Valbuena-Carabaña et al., 2010). However, climatic conditions changed strikingly with the onset of Early Medieval Cold Episode (cal AD 450-950), marked by lower temperatures and greater aridity (Desprat et al., 2003; MartínPuertas et al., 2008)”*. They also state that *“The Visigothic Period was a phase of large deforestation processes (López-Sáez et al., 2009b; Abel-Schaad, 2012)... This increase is likely related to grazing activities (high values of anthropozoogenous taxa and coprophilous fungi), in order to preserve upland pastures, but also to bring new lands into cultivation, in a new expansive dynamic of the farming strategies (Blanco-González et al., 2009). In this sense, both climatic conditions and the livestock intensification prevent a further expansion of crops in mountain areas”*. According to this article, the decrease in tree cover and the increase in charcoal accumulation rates during the Visigothic period could be related both to climatic changes and to grazing activities and development of new crops, in an expansive dynamic of livestock strategies (Blanco-González et al., 2009; López-Sáez et al., 2014). In addition, López-Sáez et al. (2014) also mention that rye (*S. cereale*), a cereal better adapted to low temperatures, appears in several sites of the Central System during this period, pointing to the aforementioned increasing human pressure on mountain areas. This displacement of human population to higher elevations, as we have also mentioned in the manuscript, could be related to more arid conditions and severe droughts that pushed human populations to higher altitudes. However, as the reviewer suggested, we have included a new

sentence that clarifies the possibility of human influence on the vegetation, even if it could also have been a consequence of climatic changes and droughts (Lines 185-198).

It must be noted that the study is highly dependent on pollen data, although there are other records that provide information on the climate of the period under analysis. The authors are invited to consider whether to include some references to these studies in the article. I refer especially to the analysis of variations in mercury concentrations in the Penido Vello deposits (Martínez-Cortizas et al., 1999), as well as to the investigation of some marine sediments. Specifically, data obtained in marine deposits from the Tagus estuary (Abrantes et al., 2005; Rodrigues et al., 2009), from the Vigo estuary (Álvarez et al., 2005) and from the western Mediterranean basin (Nieto-Moreno et al., 2011, cited by the authors) are relevant.

The paper from Martínez-Cortizas et al. (1999) is interesting when interpreting temperature climate conditions based on the presence of Hg with different thermal stabilities, but it is not that strong for paleo-precipitation conditions. In addition, the sample-resolution is not that high to observe environmental and climate changes (in particular drought periods) recorded between the 5th to 10th centuries CE (the period of interest of our study).

Regarding the papers by Abrantes et al. (2005) and Rodrigues et al. (2009), although the changes in temperatures recorded in the papers have been related to variations in the NAO, these conclusions are mainly based on the correlation between temperatures recorded by marine proxies related to SST (diatoms, foraminifera, alkenones...) and their correlation with variations in precipitation observed in other papers. This paper is of great relevance with respect to temperature changes recorded in marine environments, but it is not that relevant as a proxy for terrestrial climate and thus not ideal for a comparison with high-resolution continental precipitation records, such as our *Artemisia* and xerophyte stacks or the $\delta^{18}\text{O}$ and $\delta^{13}\text{C}$ records from the Chaara (Morocco) and Cobre-Caite-Mayor (Spain) caves that we have included in Figure 2 and in discussion. The same applies to the article by Álvarez et al. (2005), in which the combined study of coccolith assemblages and variations in biomarkers is of great impact for suggesting environmental changes in temperatures but is not robust enough for observing changes in precipitation.

With respect to the two Alboran marine records from Nieto-Moreno et al. (2011) that we already had cited, they present relatively low-resolution records. Nevertheless, we thought it was interesting to include this citation in the main text because they show both fluvial and aeolian inputs, which are closely related to aeolian flows from the Sahara and arid conditions. However, the resolution is not high enough to include it in Figure 2 in comparison with the high-resolution *Artemisia* and xerophyte stacks shown in this paper or the isotopic records from the Spanish and Moroccan caves. Having said that, and due to the high quality of the cave records that we already included in the figures together with the comparisons of environmental records that we pointed out in the main text (e.g., Nieto-Moreno et al. 2011), we consider that it is not necessary to include more paleoenvironmental/paleoclimate papers such as those by Martínez-Cortizas et al. (1999), Abrantes et al. (2005), Rodrigues et al. (2009) and Álvarez et al. (2005) suggested by the reviewer.

The presentation of archaeological data is very brief (it is accepted that this is not the aim of the work). However, the contribution by Martín-Viso used as the main reference (Martín Viso, 2013) deals with a specific problem of regional scope (the archaeological contexts of the 'Visigothic slates'). This same author has published another study much more closely related to the problem to be addressed (Martín Viso, 2012). On the other hand, it is not true that archaeological data referring to the Visigothic period are scarce. An update of the very abundant archaeological

information on the period can be found in a recent monograph (Diarte-Blasco, 2018). On the archaeology of Islam in the Iberian Peninsula, there are also syntheses to be considered (Manzano, 2006: 238-216).

We acknowledge the reviewer comment. According to our data and other interpretations of climate and archaeological data/results shown in the text, there was a clear environmental and climate change during the identified droughts periods (subjected to the effect of human activity on vegetation, as shown in a comment above and in the new version of the manuscript). These conditions, in one way or another, should have affected the development of production strategies (Lines 141-147, 185-198 and 230-255). Although the pollen results at the end of the Visigothic period is characterized by the drought period II between 695 and 725 CE, it is difficult to determine if this climate anomaly was a determining factor in the political situation that arose from 711 CE onwards. In addition, the cultivation in al-Andalus was based on the Mediterranean triad of crops (cereals, olives and vines) and it is clear that the Andalusian society intensified agricultural strategies capable of overcoming the limitations imposed by the climate (Barceló et al., 1996) (Lines 281-287). However, with the recent archaeological data, we cannot determine whether there was a notable variation in the agricultural stocks between the Visigothic Kingdom and al-Andalus.

The extensive monograph by Diarte-Blasco (2018), mentioned by the reviewer, presents a large amount of archaeological information about the Visigothic period. However, we have found no evidence of archaeological remains for the last phase of the reign (end of the 7th century - beginning of the 8th century) during which the drought periods that we identify with the Iberian-Moroccan pollen stacks occurred. In this monograph, when the author talks about “The demise of Visigothic Hispania”, she says that “*Unfortunately, our information about these last phases of the kingdom is poor, because after 694 we lack further acts of the Councils (we do know that, during Wittiza’s reign, a Church Council was held in Toledo, but its acts are not preserved); the so-called Chronicle of 754 or Mozarabic Chronicle, composed towards the end of the 8th century, when Spain was already under the Arab rule, is probably one of the best sources for this period.*”. According to this, apart from the Mozarabic chronicle of 754 that we have already used in the manuscript as one of the best historical sources for this period, there seems to be no evidence of infrastructures adapted to the periods of drought that occurred during this final phase, which prevents the scientific community from giving an inflexible opinion and having a unanimous position on what was the main cause of the fall of the kingdom, although it is most likely that it was not a single factor. Nevertheless, we found this book extremely interesting and one of the reference books/articles on the post-Roman period between the 5th and 8th centuries. The author also contextualizes this post-Roman period with sections about archaeology and historical sources of the Roman period, which had great importance, for example, in urban, political and agricultural aspects for the Visigothic state. Therefore, we have cited this book and included a new sentence in the Introduction section (Lines 46-50).

It is not entirely certain that the Visigothic kingdom faced a serious political and administrative crisis before the Islamic conquest. This traditional reading is being challenged by more recent research, which has questioned the decadence, which has been attributed to it in the 7th century and emphasised instead the dynamic and reformist character of the Visigothic state in the years before its demise (Díaz, 2012).

Thanks for the comment and the reference. There are different hypotheses about the decline of the Visigothic state. During the last two decades, it has been argued that the political crisis that the Visigothic kingdom was facing did not necessarily lead to its final decline. However, recent studies underline the inopportune arrival of Muslim troops during the conflict between the different

factions of the Visigothic kingdom (Díaz, 2012). In addition, the long conflict between the heirs of Witiza and King Rodrigo did not result in a civil war, but it may have intensified after the severe droughts at 706-710 CE reported by the historical sources (Sénac, 2021) and identified in our pollen results at 695-725 CE (Fig. 3). We have included some sentences about these questions in Lines 214-221.

In lines 161-162: The Visigoths tried to adapt to such drought conditions in the 7th and 8th centuries CE using different techniques. The fact is that the Visigothic kingdom disappeared in 711, so this chronological framework needs to be better defined.

We have modified this sentence. See Lines 191-192.

The contribution is judged to be consistent in its presentation of the data (although improvements are suggested). The fact that I do not share the authors' point of view on the determinant character of climatic changes in the crisis of the early medieval kingdoms should not be considered an obstacle to its publication.

Thank you for the comment. We really appreciate your comments/suggestions and constructive feedback. We hope that you find the modifications and the new version of the article satisfactory. Although you feel that the article has a climatic deterministic character, we have tried to soften that message and not state that the identified droughts were the only factor in the fall of the Visigothic Kingdom and in other historical events between the 5th and 10th centuries CE. In this way, we believe that this new version is much more robust from an archaeological point of view, and in the interpretation of the drought periods and their possible consequences in the described historical events, which were essential in the cultural, political and religious situation of the Iberian Peninsula and part of the Mediterranean during the following centuries.

References:

- Abrantes, F., Lebreiro, S., Rodrigues, T., Gil, I., Bartels-Jónsdóttir, H., Oliveira, P., Kissel, C & Grimalt, J.O. *Shallow-marine sediment cores record climate variability and earthquake activity off Lisbon (Portugal) for the last 2000 years. Quaternary Science Reviews* 24, 2477-3494 (2005).
- Álvarez, M.C., Flores, J.A., Sierro, F.J., Diz, P., Francés, G., Pelejero, C. & Grimalt, J. *Millennial surface water dynamics in the Ría de Vigo during the last 3000 years as revealed by coccoliths and molecular biomarkers. Palaeogeography, Palaeoclimatology, Palaeoecology* 218, 1-13 (2005).
- Cheyette, F.L. *The disappearance of the ancient landscape and the climatic anomaly of the early Middle Ages: a question to be pursued. Early Medieval Europe* 16(2), 127-165 (2008).
- Diarte-Blasco, P. *Late Antique and Early Medieval Hispania. Landscapes without Strategy. An archaeological approach (Oxbow Books, 2018).*
- Díaz, P.C. *La dinámica del poder y la defensa del territorio: para una comprensión del fin del reino visigodo de Toledo, In Actas de la XXXIX semana de estudios medievales. [De Mahoma a Carlomagno. Los primeros tiempos (siglos VII-IX), 168-205 (2012).*
- Harper, K. *The Fate of Rome. Climate, Disease and the End of an Empire (Princeton University Press, 2017).*
- Jalut, G., Dedoubat, J.-J., Fontugne, M., Otto, T. *Holocene circum-Mediterranean vegetation changes: Climate forcing and human impact. Quaternary International* 200, 4-18 (2009).

López-Sáez, J.A., Abel-Schaad, D., Pérez-Díaz, S., Blanco-González, A., Alba-Sánchez, F., Dorado, M., Ruiz-Zapata, B., Gil-García, M.J., Gómez-González, C. & Franco-Música, F. *Vegetation history, climate and human impact in the Spanish Central System over the last 9000 years*. *Quaternary International* 353, 98-122 (2014).

Manzano, E. *Conquistadores, emires y califas. Los Omeyas y la formación de al-Andalus* (Crítica, 2006).

Martín Viso, I. *The “Visigothic” slates and their archaeological contexts*. *Journal of Medieval Iberian Studies* 5, 145-168 (2013).

Martín Viso, I. *Un mundo en transformación: los espacios rurales en la Hispania post-romana (siglos V-VII)*. In *Visigodos y Omeyas. El territorio* (ed L. Caballero, P. Mateos & T. Cordero) (Consejo Superior de Investigaciones Científicas, 2012) 31-63.

Martínez-Cortizas, A., Pontevedra-Pombal, X., García-Rodeja, E., Nóvoa-Muñoz, J.C. & Shotyk, W. *Mercury in a Spanish peat bog: archive of climate change and atmospheric metal deposition*. *Science* 284, 939-942 (1999).

McCormick, M., Büntgen, U., Cane, M.A., Cook, E.R., Harper, K., Huybers, P., Litt, T., Manning, S.W., Mayewski, P.A. More, A.F.M. Nicolussi, K. & Tegel, W. *Climate Change during and after the Roman Empire: Reconstructing the Past from Scientific and Historical Evidence*. *Journal of Interdisciplinary History* 43(2), 169-220 (2012).

Nieto-Moreno, V., Martínez-Ruiz, F., Giral, S., Jiménez-Espejo, F., Gallego-Torres, D., Rodrigo-Gámiz, M., García-Orellana, J., Ortega-Huertas, M. & de Lange, J.G. *Tracking climate variability in the western Mediterranean during the Late Holocene: a multiproxy approach*. *Climate of the Past* 7, 1395-1414 (2011).

Rodrigues, T., Grimalt, J.O., Abrantes, F.G., Flores, J.A. and Lebreiro, S.M. *Holocene interdependences of changes in sea surface temperature, productivity, and fluvial inputs in the Iberian continental shelf (Tagus mud patch)*. *Geochemistry, Geophysics, Geosystems* 10(7), 1-17 (2009).

Walsh, K. *Risk and marginality at high altitudes: new interpretations from fieldwork on the Faravel Plateau, Hautes-Alpes*. *Antiquity* 79, 289-305 (2005).

References:

Watson, A. M. *The Arab agricultural revolution and its diffusion, 700–1100*. *The Journal of Economic History* 34, 8-35 (1974).

Blanco-González, A., López-Sáez, J.A., López-Merino, L. *Ocupación y uso del territorio en el sector centromeridional de la cuenca del Duero entre la Antigüedad y la Alta Edad Media (siglos I-XI D.C.)*. *Archivo Español de Arqueología* 82, 275e300 (2009).

López-Sáez, J.A., Abel-Schaad, D., Pérez-Díaz, S., Blanco-González, A., Alba-Sánchez, F., Dorado, M., Ruiz-Zapata, B., Gil-García, M.J., Gómez-González, C. & Franco-Música, F. *Vegetation history, climate and human impact in the Spanish Central System over the last 9000 years*. *Quaternary International* 353, 98-122 (2014).

Barceló, M. et al. *El agua que no duerme: fundamentos de la arqueología hidráulica andalusí*. Granada: Sierra Nevada 95 El Legado Andalusi (1996).

Sénac, P. *Al-Andalus (Siglos VIII-XI)* (transl. Peinado-Santaella, R. G. & Zapata-Cano, P. H.) (Editorial Universidad de Granada, Granada, 2021).

Reviewers' Comments:

Reviewer #4:

Remarks to the Author:

The article has been significantly improved. The thesis that droughts could explain the disappearance of the Visigothic kingdom has been presented in a more balanced and objective way.

The authors have chosen not to include the suggested references to the climate change that characterises the DACP on the grounds that they are not relevant, given that the research makes drought episodes the object of its analysis. In my opinion, it is not possible to study drought episodes separately from the set of climate changes that characterise the DACP. Especially since the fossil pollen record reflects both climatic factors (including droughts) and the human agent. It should be noted that the text repeatedly refers to climatic changes in the historical sequence. For its characterisation it attaches relevant value to written sources that mention climatic events in the study period, although the information that has been transmitted to us probably reflects not only climatic events, but also the own (subjective and perhaps self-interested) perception of the authors who relate these events. Despite these observations, it is judged that the documentation provided on changes in the vegetal landscape of the Iberian Peninsula has been thoroughly compiled and that it is up to the authors to incorporate other data on paleoclimate into their study. I recommend the acceptance of this article on its present form in the journal 'Nature Communications.'

The reviewers' comments are in **black** and our responses in **blue**.

In this file, when we reference lines (e.g., "Lines 35-38"), we refer to the lines of the clean manuscript file "*Manuscript Camuera et al_NCOMMS-22-45602D_August2023_MainText*", which does not have the *Track Changes* activated. We have also uploaded the file with *Track changes* activated, which includes all changes performed in this revision (file "*TrackChanges_Manuscript Camuera et al_NCOMMS-22-45602D_August2023*"). In addition, the Supplementary Figures 1, 2 and 3 have been included separately in the new Supplementary Information file. The Supplementary Data 1 and 2 are provided separately in Excel documents (.xlsx file) and have also been uploaded to the Pangaea data repository.

REVIEWERS' COMMENTS

Reviewer #4 (Remarks to the Author):

The article has been significantly improved. The thesis that droughts could explain the disappearance of the Visigothic kingdom has been presented in a more balanced and objective way.

The authors have chosen not to include the suggested references to the climate change that characterises the DACP on the grounds that they are not relevant, given that the research makes drought episodes the object of its analysis. In my opinion, it is not possible to study drought episodes separately from the set of climate changes that characterise the DACP. Especially since the fossil pollen record reflects both climatic factors (including droughts) and the human agent. It should be noted that the text repeatedly refers to climatic changes in the historical sequence. For its characterisation it attaches relevant value to written sources that mention climatic events in the study period, although the information that has been transmitted to us probably reflects not only climatic events, but also the own (subjective and perhaps self-interested) perception of the authors who relate these events. Despite these observations, it is judged that the documentation provided on changes in the vegetal landscape of the Iberian Peninsula has been thoroughly compiled and that it is up to the authors to incorporate other data on paleoclimate into their study. I recommend the acceptance of this article on its present form in the journal 'Nature Communications.'

Thanks for the comment. We realized that the study by Cheyette (2008) suggested by the reviewer could be useful for its inclusion in the Introduction section. Therefore, we have included in Lines 34-37: "*Nowadays, climate is becoming a central element in explaining historical crises in the Mediterranean area, as has occurred in recent decades around the Dark Age Cold Period and its relationship with the appearance of the Justinian's plague (Little, 2005), or with the crisis that affected rural settlements in some parts of Europe during those centuries (Cheyette, 2008).*".

We agree with the reviewer about the limitations of the pollen analysis and we have therefore included a new paragraph in Methods (Lines 356-363) referring to this limitation: "*The limitations of the pollen analysis in paleoclimate interpretations have to be taken into account, since modifications in the vegetation cover caused by human impact could also have affected pollen abundances. However, it is more likely that the human effect on vegetation could have had an impact during a given period in a particular area of the Iberian Peninsula and/or Morocco, but it is less likely that this effect would have affected different large Iberian/Moroccan areas at the*

same time. Therefore, in order to reduce the human effect on vegetation, we have collected a large number of pollen records over a large territory, reducing the noise of regional human disturbance that could have affected some pollen records during specific periods.”. To verify and corroborate whether or not the pollen-based drought signals are affected by anthropogenic activities, our pollen data have been compared with regional speleothems records that are not or less affected by human activity. In the main text, we have also included: *“Comparison of pollen data with isotopic analysis of speleothems provides a more detailed insight on local environmental and regional paleoclimate changes in the study region^{15,16}. In addition, since human influence on speleothem records is lower than on the vegetation records, the comparison between these two proxies allows us to corroborate whether the aridity periods observed in pollen records are predominantly natural or not.”* (Lines 72-76). The high correlation between pollen and cave records indicates that the described drought periods are mainly of natural origin.

In addition, as the reviewer mentioned, the credibility of the short-scale droughts recorded by the pollen data between 5th and 10th centuries CE are also supported by the archaeological and historical sources included in the manuscript, which indicate dry climate conditions, famines, changes in agricultural activities, population movements or the construction of new structures adapted to water shortages, among others.